



Geoscientific
Model Development

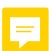

# MP CBM-Z V1.0: design for a new Carbon Bond Mechanism Z (CBM-Z) gas-phase chemical mechanism architecture for next-generation processors

**Hui Wang[1], Junmin Lin[2,a], Qizhong Wu[1], Huansheng Chen[3], Xiao Tang[3], Zifa Wang[3], Xueshun Chen[3], Huaqiong Cheng[1], and Lanning Wang[1]**

[1]College of Global Change and Earth System Science, Joint Center for Global Changes Studies,
Beijing Normal University, Beijing 100875, China
[2]Intel (China) Corporation, Beijing 100013, China
[3]State Key Laboratory of Atmospheric Boundary Layer Physics and Atmospheric Chemistry,
Institute of Atmospheric Physics, Chinese Academy of Sciences, Beijing 100029, China
[a]now at: Artificial Intelligence Research Department, JD Corp., Beijing 100101, China

**Correspondence:** Qizhong Wu (wqizhong@bnu.edu.cn) and Huansheng Chen (chenhuansheng@mail.iap.ac.cn)

**Abstract.** Precise and rapid air quality simulations and forecasting are limited by the computational performance of the air quality model used, and the gas-phase chemistry module is the most time-consuming function in the air quality model. In this study, we designed a new framework for the widely used the Carbon Bond Mechanism Z (CBM-Z) gas-phase chemical kinetics kernel to adapt the single-instruction, multiple-data (SIMD) technology in next-generation processors to improve its calculation performance. The optimization implements the fine-grain level parallelization of CBM-Z by improving its vectorization ability. Through constructing loops and integrating the main branches, e.g., diverse chemistry sub-schemes, multiple spatial points in the model can be operated simultaneously on vector processing units (VPUs). Two generation CPUs – Intel Xeon E5-2680 V4 CPU and Intel Xeon Gold 6132 – and Intel Xeon Phi 7250 Knights Landing (KNL) are used as the benchmark processors. The validation of the CBM-Z module outputs indicates that the relative bias reaches a maximum of 0.025 % after 10 h integration with *-fp-model fast = 1*CE1 compile flag. The results of the module test show that the Multiple-Points CBM-Z (MP CBM-Z) resulted in 5.16× and 8.97× speedup on a single core of Intel Xeon E5-2680 V4 and Intel Xeon Gold 6132 CPUs, respectively, and KNL had a speedup of 3.69× compared with the performance of CBM-Z on the Intel Xeon E5-2680 V4 platform. For the single-node tests, the speedup on the two generation CPUs can reach 104.63× and 198.50× using message passing interface (MPI) and 101.02× and 194.60× using OpenMP, and the speedup on the KNL node can reach 194.60×TS1 using MPI and 167.45× using OpenMP. The speedup of the optimized CBM-Z is approximately 40 % higher on a one-socket KNL platform than on a two-socket Broadwell platform and about 13 %–16 % lower than on a two-socket Skylake platform. We also tested a three-dimensional chemistry transport model (CTM) named Nested Air Quality Prediction Model System (NAQPMS) equipped with the MP CBM-Z. The tests illustrate an obvious improvement on the performance for the CTM after adopting the MP CBM-Z. The results show that the MP CBM-Z leads to a speedup of 3.32 and 1.96 for the gas-phase chemistry module and the CTM on the Intel Xeon E5-2680 platform. Moreover, on the new Intel Xeon Gold 6132 platform, the MP CBM-Z gains 4.90× and 2.22× speedups for the gas-phase chemistry module and the whole CTM. For the KNL, the MP CBM-Z enables a 3.52× speedup for the gas-phase chemistry module, but the whole model lost 24.10 % performance compared to the CPU platform due to the poor performance of other modules. In addition, since this optimization seeks to improve the utilization of the VPU, the model is more suitable for the new generation processors adopting the more advanced SIMD technology. The results of our tests already show that the benefit

of updating CPU improved by about 47 % by using the MP CBM-Z since the optimized code has better adaptability for the new hardware. This work improves the performance of the CBM-Z chemical kinetics kernel as well as the calculation efficiency of the air quality model, which can directly improve the practical value of the air quality model in scientific simulations and routine forecasting.

## 1  Introduction

Air pollution and its impacts on human health have attracted widespread attention all over the world, especially in developing countries (Gurjar et al., 2016; Zhang et al., 2017). As a useful tool for air quality problems, chemistry transport models (CTMs), are widely used in studies of air quality (Gao et al., 2016; Chen et al., 2015; Wu et al., 2014) and in establishing air quality forecasting (AQF) systems. As the core of the AQF system, a CTM requires a large number of computational resources to simulate the complex chemical and physical processes. To satisfy the demand of routine air quality forecasting in a timely manner, coarse spatial resolution and relatively simple processes are adopted in CTMs to minimize the use of computational resources. Meanwhile, other simulation studies with more complex processes are also limited by computational resources. Therefore, air quality studies can benefit significantly by improving the performance of the CTM used.

In a CTM, the most time-consuming module is the gas-phase chemistry module (Wang et al., 2017). The gas-phase chemistry module is described as a system of ordinary differential equations (ODEs) to simulate the chemical kinetics of trace gases in an atmosphere model (Seinfeld and Pandis, 2012). Linford et al. (2009) reported that the Regional Acid Deposition Model version 2 (RADM2) (Zimmermann and Poppe, 1994; Chang et al., 1987), a chemical kinetics kernel, accounted for 90 % of the computational time in the Weather Forecasting and Research/Chemistry (WRF-chem) model (Grell et al., 2005). Another widely used chemical kinetics kernel, the Carbon Bond Mechanism version Z (CBM-Z) (Zaveri and Peters, 1999), accounts for approximately 68 % of the computation time in the Global Nested Air Quality Prediction Model System (GNAQPMS) (Chen et al., 2015; Wang et al., 2017). Therefore, accelerating the gas-phase chemistry module can directly improve the performance of the CTM as well as the whole AQF system. The AQF system can also benefit from the performance improvement by adopting a higher model resolution and improving the frequency of air quality forecasting.

The performance of models improves with updated hardware. However, by reaching the bottleneck of power density and the thermal limitation of the silicon technology for a single-core design, frequent updating has not been an efficient way to improve the scientific model's performance. Additionally, multicore architecture and a heterogeneous computing architecture such as a Many Integrated Core (MIC) and a graphic processing unit (GPU) have become the hardware trend for high-performance computing (Xu et al., 2015; Lawrence et al., 2018). Meanwhile, to take full advantage of the advanced features of new processor architecture, the applications or the models must be redesigned or rewritten. Xu et al. (2015) rewrote the Princeton Ocean Model (POM) using Compute Unified Device Architecture-C (CUDA-C) to port it from a CPU to a GPU platform. Linford et al. (2009) also tried to solve the computation bottleneck of RADM2 mentioned above by using a heterogeneous platform such as GPU–CPU. In addition, our previous work showed the primary optimizations we performed to accelerate the GNAQPMS on the new generation CPU and Intel MIC platforms (Knights Landing, KNL; Sodani et al., 2016) and had a significant performance improvement on both platforms, a 2.77× speedup on CPU and a 3.51× speedup on the KNL node (Wang et al., 2017). In this study, we redesign the code structure of the chemical kinetics kernel CBM-Z to improve its vectorization performance on the CPU and KNL platforms, which significantly improves its performance by fully utilizing the single-instruction, multiple-data (SIMD) technology. We tested the performance of this optimized CBM-Z module as well as a regional CTM equipped with it. The code test only contained this single module, making it easier to let the CTM developers reuse the code.

Section 2.1 in this paper introduces the CBM-Z scheme, and Sect. 2.2 describes the new architecture we designed for CBM-Z. Since multiple spatial points were operated simultaneously in the optimized CBM-Z scheme, the optimized CBM-Z scheme was called the Multiple-Points CBM-Z Version 1.0 (MP CBM-Z V1.0). In Sect. 3.1, we present our benchmark platforms. In Sect. 3.2 and 3.3, we introduce the test cases and present the test results of single-model tests and CTM tests separately. The conclusions and discussions are given in Sect. 4.

## 2  Method description

CBM-Z is a core module in CTMs that simulates the complex gas-phase chemical processes in the atmosphere. In this module, too many options and poor load balancing within the model grid boxes make it a challenge to improve its performance on a vectorization level. This leads to poor performance of CBM-Z on the new generation processors that are highly dependent on powerful vector processing units (VPUs). In our previous work, we conducted several optimizations on CBM-Z to enhance its vectorization and parallel performance (Wang et al., 2017). In this work, we attempt to further enhance its vector calculation ability by constructing a new structure, which makes the CBM-Z module suitable to be vectorized. The CBM-Z module was extracted as

an individual box model to test its performance and improve code reusability.

## 2.1 Description of CBM-Z

CBM-Z is a lumped-structure photochemical mechanism
that was developed to meet the needs of city-scale to global-scale tropospheric chemical simulations (Zaveri and Peters, 1999). The original scheme contains 67 species and 132 reactions. CBM-Z has been widely used in CTMs, e.g., the WRF-Chem (San José et al., 2015), the Nested Air Quality Pre-
diction Model System (NAQPMS) (Wang et al., 2001) and the GNAQPMS. In the NAQPMS and GNAQPMSs, CBM-Z was further modified by Li et al. (2012). It was updated to 76 species, and 28 heterogeneous reactions were added. The CBM-Z solver uses the modified backward Euler (MBE)
solver developed by Feng et al. (2015), a faster and more robust algorithm which overcomes inflexibility and preserves the non-negativity.

The main control flow of CBM-Z is shown in Fig. 1. The *IntegrateChemistry* function is treated as the core function
of the module. CBM-Z contains five chemistry sub-schemes. They are the Common Chemistry Scheme (COM), the Urban Chemistry Scheme (URB), the Biogenic Chemistry Scheme (BIO), the Marine Chemistry Scheme (MAR), and the Heterogeneous Chemistry Scheme (HET). The integration of
25 different sub-schemes is used to satisfy the simulation of diverse scenarios and scales. The combination of sub-schemes relies on the concentration and emission of each chemical species in the specific model grid, which is implemented in the *SelectGasRegime* function. The variable *iregime* stores
the return-value of *SelectGasRegime* and controls the subsequent calculation processes of CBM-Z. The possible values and the sub-schemes represented are shown in Table 1. The combinations include the COM and HET schemes, while other schemes are added when the concentration or emission
of a corresponding species in a certain scheme are greater than zero. Compared with the algorithm computing all chemical interactions, this algorithm is helpful in saving computational resources on a simple core, while such irregular and unbalanced calculations lack well-structured loops
and impede the vectorization of code. Besides the chemistry sub-schemes mentioned above, CBM-Z uses other functional branches, e.g., nocturnal and diurnal chemistry, and they impede the vectorization of the computation. The CBM-Z also contains multiple unconstructed scalar operations. We
partially integrated the scalar operations by using indirect indexing to construct loops for vectorization (Wang et al., 2017). However, this method required significant effort, and it only reconstructed a limited number of scalar operations. The CBM-Z module still contains many scalar operations.
With multi-level control flow divergences and many scalar calculations, it is not feasible to perform automatic vectorization with an Intel compiler.

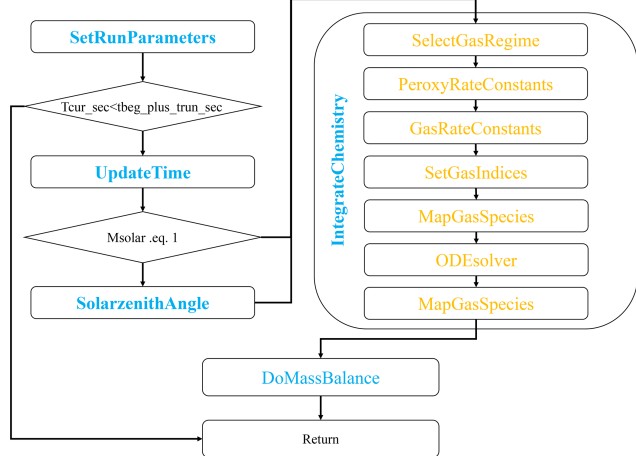

**Figure 1.** The framework of the CBM-Z gas-phase chemistry module. The functions in the yellow font represent the inner function of *IntegrateChemistry*.

**Table 1.** The possible values of *iregime* and the combination of chemical schemes.

| *iregime* | 1 | 2 | 3 | 4 | 5 | 6 |
|---|---|---|---|---|---|---|
| Sub-schemes | COM HET | COM HET URB | COM HET URB BIO | COM HET | COM HET URB | COM HET URB BIO |
| | | | | MAR | MAR | MAR |

Fortunately, contiguous model grid boxes may have similar chemical processes in air quality simulations, which provides the opportunity to integrate the grid boxes with similar 55 or the same chemical processes to implement vectorization to calculate the processes of multiple grid boxes simultaneously. The following section introduces the details about integrating the chemistry sub-schemes to implement the vectorization. 60

## 2.2 Algorithm description

The new generation Intel CPU (e.g., Skylake) and Intel MIC chips are equipped with the AVX-512 (AVX – Advanced Vector Extensions) or more advanced vectorization instructions, which support a maximum of 8 double-precision and 65 16 single-precision operations with 512 bit wide vector registers. It is critical to peak performance of the next-generation CPUs and MICs to fully reach the potential of the AVX-512 (Mielikainen et al., 2014). As mentioned in Sect. 2.1, automatic vectorization using a compiler is impeded by the 70 features of CBM-Z, and the common manual measures including constructing loops, avoiding the loop–data dependence, and aligning the data with directives are needed to further vectorize CBM-Z. On the other hand, to implement the vectorization of the module, the general design allowed 75

the CBM-Z module to handle multiple grid boxes in one citing cycle, and the functions in CBM-Z were reconstructed by adding a regular loop for these grid boxes. Subsequently, these loops can be vectorized to implement the fine-grained parallelization on a VPU.

All of the model grid boxes are distributed to multiple cores using a message passing interface (MPI) and OpenMP, which is a type of coarse-grain parallelization. Our goal is to implement fine-grained parallelization based on the SIMD, and the grid boxes that are distributed to a specific processor operate in parallel using the VPUs on each core. As shown in Fig. 2, the calling method of the CBM-Z module changes from calculating one model grid box calculation at a time to multiple model grid boxes at the same time. The step length (VLEN in Fig. 2) of the loops represents the number of the grid boxes operated simultaneously, and it is determined by the length of the vector register. The VLEN was set to 16 since the 512 bit wide vector of the AVX-512 can support 16 single-precision operations at the same time. Using this framework, the functions in CBM-Z construct an extra loop to manage the point number dimension, and the corresponding variables require an extra dimension to store the information of multiple grid boxes. Using the structure with an extra loop, it was easier to implement vectorization. Meanwhile, to avoid multiple remaining points which cannot satisfy the VLEN, we set a common variable array, *pmask* (VLEN) as shown in Fig. 2, to store the availability label of the model grid boxes. When the number of remaining grid boxes did not reach VLEN, the corresponding *pmask* value of excessive grid boxes was set to "false" to mask these grid boxes in the calculation. Furthermore, the latitude and longitude dimension loops were merged, from nested loops to a single loop, to reduce the number of unavailable points as shown in Fig. 2. Achieving such a large-scale vectorization also requires load balancing of the calculation processes, but the calculation branches in CBM-Z are an obstacle to this. Therefore, the branches in CBM-Z should be taken into consideration in constructing the loops, especially the chemical schemes chosen in Table 1. As mentioned in Sect. 2.1, the contiguous model grid boxes may have similar chemical processes in the atmosphere. This provides an opportunity to integrate the sub-schemes by masking the heterogeneous model grid boxes, and this type of masking operation can be used in the functions *GasRateConstants* and *ODEsolver* (Fig. 1). Figure 3 shows the flowchart for masking the model grid boxes to satisfy the vectorization of the grid array. A set of grid boxes with the number of VLEN (16 in this study) would perform the operation simultaneously, and the variable *pmask* signed the valid grid boxes. Meanwhile, the variable *iregime* described in Table 1 and representing the combination of sub-schemes, is used to determine whether the model grid must perform the subsequent operation or not. The grid boxes with the same property or calculation are kept by setting the variable *bmask* to "true". The COM and HET schemes are common for all grid boxes, and the

mask operation for COM and HET schemes only determines the availability of the grid boxes. As shown in Fig. 3, for the URB, BIO, and MAR schemes, the *iregime* value and *pmask* are both used to filter the heterogeneous grid boxes and the *bmask* stores the results. To improve the efficiency of vectorization, the *bmask* does not prevent the calculation of heterogeneous grid boxes but prevents the calculation results from being copied back to the return value. Thus, all computations are performed on all grid boxes, but only the results of the valid grid boxes are returned. This improves the utilization of data as well as the efficiency of vectorization. Because of the independence of the grid boxes, the computation process of VLEN arrays is independent and satisfies the requirement of vectorization, and the corresponding directives were added to declare the independence of the arrays and force the compiler to perform the data alignment and vectorization after the reconstruction of the code. Overall, by constructing the loops, the computations of the independent grid boxes were integrated with the fine-level parallel implementation through the SIMD. In addition, the efficiency of such algorithms is linearly improved with the development of the width of the vector in the VPU.

## 3  Test results

The validation and evaluation of the improvement of the new method were conducted using the box model of CBM-Z as well as a regional CTM named Nested Air Quality Prediction Model System with the optimized CBM-Z scheme. We tested the theoretical performance of vectorization by using the box model, and the CTM tests illustrate its potential in three dimensions with varying chemical regimes.

### 3.1  Benchmark platform description

The computation cluster for tests was provided by the Institute of Atmospheric Physic (IAP), Chinese Academy of Sciences (CAS). The CPU and KNL platforms were used for testing the code. The CPU platforms in this study include two generation CPUs, two-socket CPU nodes with Broadwell architecture 2.4 GHz 14-core Intel Xeon E5-2680 V4 processors, and two-socket CPU nodes with 2.6 GHz Skylake architecture 14-core Intel Xeon Gold 6132. To the vector instructions, the previous generation of Broadwell adopted the AVX-2 vector instructions and the new generation used the AVX-512 vector instructions. The AVX-512 and AVX-2 instructions support 16 and 8 single-precision floating-point calculations simultaneously, respectively. Comparing the two generation CPUs helped to present the potential of new MP CBM-Z to fully use the development of hardware. The KNL node contained one 1.4 GHz 68-core Intel Xeon Phi 7250 processor, which also adopted the AVX-512 vector instructions. The operating system was Cent OS Linux 7.4.1708 for all platforms. The code was all compiled us-

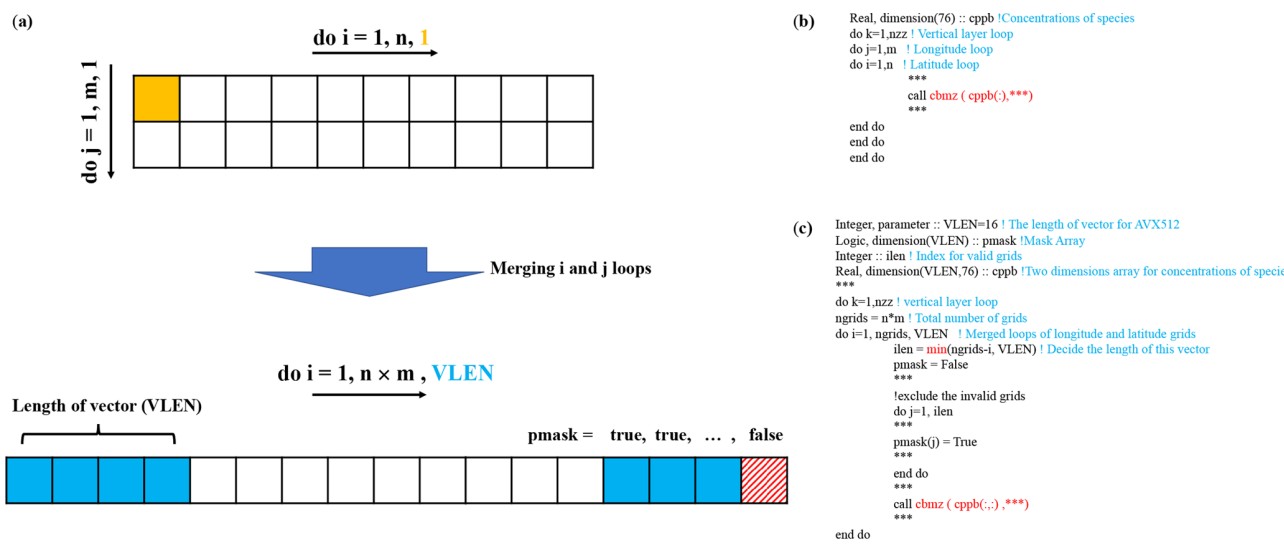

**Figure 2.** A schematic diagram of the changes in the calling method of CBM-Z. The calling method of the CBM-Z module changes from calculating one model grid calculation at a time to multiple model grid boxes at the same time. The VLEN represents the number of points operated simultaneously, which is determined by the length of the register in the vector processing unit (VPU). The $i$ and $j$ loops, equaling latitude and longitude loops, were merged to construct one vector to reduce the number of unfilled vectors. Panel **(b)** and **(c)** illustrate the sample code before and after integrating grid boxes.

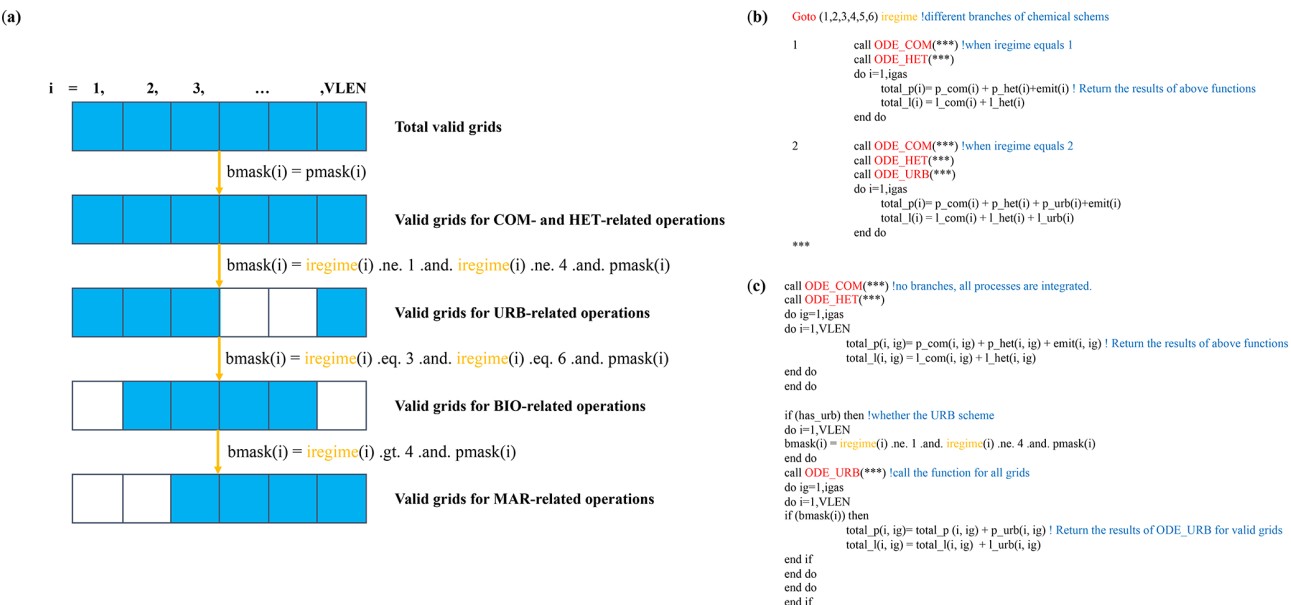

**Figure 3.** The flowchart **(a)** shows the way to mask the heterogeneous girds to integrate grid boxes to perform the vectorization operations according to the *iregime* values. Panels **(b)** and **(c)** illustrate the sample code before and after integrating grid boxes. In panel **(b)**, *iregime* leads different calling processes; in panel **(c)**, the calling processes are integrated into one flow, and the functions are called for all grid boxes but only the values of valid grid boxes are returned.

ing the Intel FORTRAN Compiler 2017 update 4, and the compile flags for vectorization and float-pointing accuracy of the CBM-Z module and the NAQPMS are shown in Tables 2 and 3, respectively. The corresponding flags for vectorization (e.g., *-xCORE-AVX2*, *-xCOMMON-AVX512*, *-xMIC-AVX512*, *-align array64byte*) were adopted for MP CBM-Z.

We also tested the code using diverse options for the compile flag *-fp-model*, which controls the balance between accuracy and performance of floating-point calculations, to investigate its impact on code. We mainly consider the two options of *-fp-model precise* and *-fp-model fast=1*. The *fast=1* is the default option when *-fp-model* flag is not selected. Compared

with the option *precise*, the *fast=1* improves the computational performance but reduces the accuracy of the floating-point calculations. Using *precise* is a safer option and forces the compiler to avoid the vectorization of some calculations to improve accuracy. We compare the results of the two options, including the outputs and the performance of models, to investigate its impact and discuss a suitable choice of compile flag.

## 3.2  Box model test

The box model of MP CBM-Z was used to validate the model outputs and investigate the ideal parallel performance of the single module. We also tested the results using different parallelization techniques, e.g., MPI and OpenMP. Each test was repeated 10 times to reduce the impact from any platform variability.

### 3.2.1  Test case description

There are two cases that were used for the CBM-Z box model. One was a 10 h single grid box case with all species to validate the outputs of the model, and the other was a 1 h simulation with $160 \times 148 \times 20$ grid boxes to test the performance of the module under a more realistic scenario. The initial values for the single grid box are shown in Table S1 in the Supplement. The meteorological conditions were constant and emissions were set to zero to test the error of the algorithms. The time step was 5 s for the two cases. For validation purposes, output every 5 min was used, while the computational performance test did not include the output function to eliminate any impact from input–output (I/O). The different compiling flags for the precision of floating-point calculations are presented in Table 2. We test the baseline and the optimized model on two different platforms of CPU and KNL, and the computational time was counted using the *system_clock* function.

### 3.2.2  Box model validation

We evaluate the chemical species including ozone ($O_3$), nitrogen dioxide ($NO_2$), nitrogen monoxide (NO), hydrogen peroxide ($H_2O_2$), sulfur dioxide ($SO_2$), sulfuric acid ($H_2SO_4$), hydroxyl (OH) radical, hydroperoxyl ($HO_2$) radical, and alkyl peroxy ($RO_2$) radical. These species are important for tropospheric gas-phase chemistry and sulfate aerosol formation and hence suitable for validating whether the optimization significantly changed the simulated results or not. Figure 4 shows the time series of the simulated concentrations of the species by the baseline (base) and the optimized (opt) model with the *precise* and *fast=1* compile flags. The results with the baseline code with *precise* compile flag is the benchmark, and there is no difference between the results from the baseline and optimized code with the same *precise* compile flag. The *precise* compile flag is a relatively safe compile flag and prohibits optimizations that can affect

the accuracy. The *fast=1* compile flag can lead to errors even with the same code, but the relative error (RE) of the baseline code with *fast=1* compile flag relative to the benchmark is extremely small ($< 0.0002$ %). As shown in Fig. 4, with the optimized code, the *fast=1* compile flag results in a maximum RE of 0.025 % for NO and $NO_2$ at the end of the simulation. We find that the error caused by the *fast=1* compile flag did not become obvious for species with low concentrations of OH and $RO_2$. We will further discuss the impact of the *fast=1* compile flag in Sect. 3.3.2 in the context of CTM simulations.

### 3.2.3  Box model computational performance

The case with $160 \times 148 \times 20$ grid boxes was used to test the computational performance. Both the baseline and the optimized version of CBM-Z contained the same 76 species. The computational time of the baseline version on a single core of E5-2680 V4 CPU with the *precise* compile flag was considered as the benchmark time. The tests were done with two generations of CPUs and KNL.

The option of *-fp-model* could directly affect the performance. As shown in Table 4, the benchmark performance was 1014.67 s on the E5-2680 V4 platform. By using the new platform with Intel Gold 6132, the baseline version code achieves $1.52 \times$ speedup with the *precise* compile flag. The *fast=1* compile flag leads to $1.28 \times$ and $2.04 \times$ speedups for the baseline code on both CPUs. Meanwhile, updating the CPU enables the original CBM-Z module to gain a speedup of about 1.52 and 1.59 with *precise* and *fast=1* compile flags, respectively.

The MP CBM-Z module shows good performance on both CPUs. On the E5-2680 V4 CPU with Broadwell architecture, the optimized code with two different compile flags consumed 581.14 and 153.32 s, respectively; meanwhile, the speedups reach $1.75 \times$ and $6.62 \times$ compared with the benchmark performance. In regard to the Intel Gold 6132 platform, the optimized version CBM-Z consumed 352.00 and 55.42 s with *precise* and *fast=1* compile flags, respectively. Compared with the benchmark time, the speedups reach $2.88 \times$ and $18.31 \times$. By using the same *fast=1* compile option, the MP CBM-Z shows $5.16 \times$ and $8.97 \times$ speedups on two generations of CPU compared with the original CBM-Z code.

The results also illustrate that the optimized code could better utilize the updating of cores through good vectorization ability compared with the baseline code. Comparing the performance of the optimized code, we find that updating the CPU could lead to about 1.65 times and 2.76 times acceleration with *precise* and *fast=1* compile flags, respectively, which is higher than the $1.5 \times$ speedup gained with the baseline code.

Compile flags largely affect the code performance on KNL. On the Xeon Phi 7250 platform, the optimized code took 3454.90 s with the *precise* compile flag since the majority of vectorizations were forbidden, and it is even slower

**Table 2.** Compile flags of the different versions of CBM-Z.

| Version of CBM-Z | Processor | Intel compiler flags | |
|---|---|---|---|
| | | Flags for vectorization | Flags for floating-point accuracy |
| Baseline CBM-Z | Xeon E5-2680 V4 | *-xCORE-AVX2*<br>*-xCORE-AVX2* | *-fp-model precise*<br>*-fp-model fast* $= 1$ |
| | Xeon Gold 6132 | *-xCOMMON-AVX512*<br>*-xCOMMON-AVX512* | *-fp-model precise*<br>*-fp-model fast* $= 1$ |
| MP CBM-Z | Xeon E5-2680 V4 | *-xCORE-AVX2*<br>*-xCORE-AVX2* | *-fp-model precise*<br>*-fp-model fast* $= 1$ |
| | Xeon Gold 6132 | *-xCOMMON-AVX512*<br>*-xCOMMON-AVX512* | *-fp-model precise*<br>*-fp-model fast* $= 1$ |
| | Xeon Phi 7250 | *-xMIC-AVX512* | *-fp-model fast* $= 1$ |

**Table 3.** Compile flags of the different versions of NAQPMS.

| Version of NAQPMS | Processor | Intel compiler flags | |
|---|---|---|---|
| | | Flags for vectorization | Flags for floating-point accuracy |
| Baseline NAQPMS | Xeon E5-2680 V4 | *-xCORE-AVX2*<br>*-xCORE-AVX2* | *-fp-model precise*<br>*-fp-model fast* $= 1$ |
| | Xeon Gold 6132 | *-xCOMMON-AVX512*<br>*-xCOMMON-AVX512* | *-fp-model precise*<br>*-fp-model fast* $= 1$ |
| NAQPMS with MP CBM-Z | Xeon E5-2680 V4 | *-xCORE-AVX2*<br>*-xCORE-AVX2* | *-fp-model precise*<br>*-fp-model fast* $= 1$ |
| | Xeon Gold 6132 | *-xCOMMON-AVX512*<br>*-xCOMMON-AVX512* | *-fp-model precise*<br>*-fp-model fast* $= 1$ |
| | Xeon Phi 7250 | *-xMIC-AVX512* | *-fp-model fast* $= 1$ |

than the benchmark performance; it only took 214.09 seconds and obtained a speedup of 4.74× with the *fast=1* compile flag. Compared with the baseline CBM-Z with the *fast=1* flag on Intel Xeon E5-2680 V4, KNL gains a 3.69× speedup with the MP CBM-Z.

In addition, the baseline and optimized code with *fast=1* were also analyzed by using the high-performance computing (HPC) performance characterization from the Intel VTune tools on the CPU platform. On the Intel Gold 6132 platform, the single-precision giga-floating point operations calculated per second (GFLOPS) increased from 4.81 to 21.37 compared with the original CBM-Z module, and the vector capacity usage improved from 14.3 % in the baseline CBM-Z to 89.4 % in the MP CBM-Z, which implies that the majority of floating-point instructions in CBM-Z were vectorized.

We also tested the parallel version of the MP CBM-Z by compiling with the *fast=1* option and with MPI and OpenMP separately. We evaluated the speedups based on the performance of the baseline CBM-Z on the Intel Xeon E5-2680 V4 platform with *fast=1* option. The results are shown in Table 5. The MPI and OpenMP version of CBM-Z had a 104.63× speedup and 101.02× speedup on the Intel Xeon E5-2680 V4 platform. On the new Intel Xeon Gold 6132, the MP CBM-Z got a speedup of 198.50× and 194.60× with MPI and OpenMP. For the KNL, the speedup reached 175.23× by using MPI and 167.45× by using OpenMP, which was approximately 40 % faster than those on the two-socket Broadwell platform with AVX2 vectorization instruction and about 13 %–16 % slower than those on the two-socket Skylake platform with the same AVX512 vectorization instruction. The combination of the fine-grain vectorization and the coarse-grain parallelization of OpenMP/MPI results in a significant performance improvement on the new generation processors. The enhancement of the vectorization performance may be the key to fully using the new generation processors equipped with advanced and wider vectors

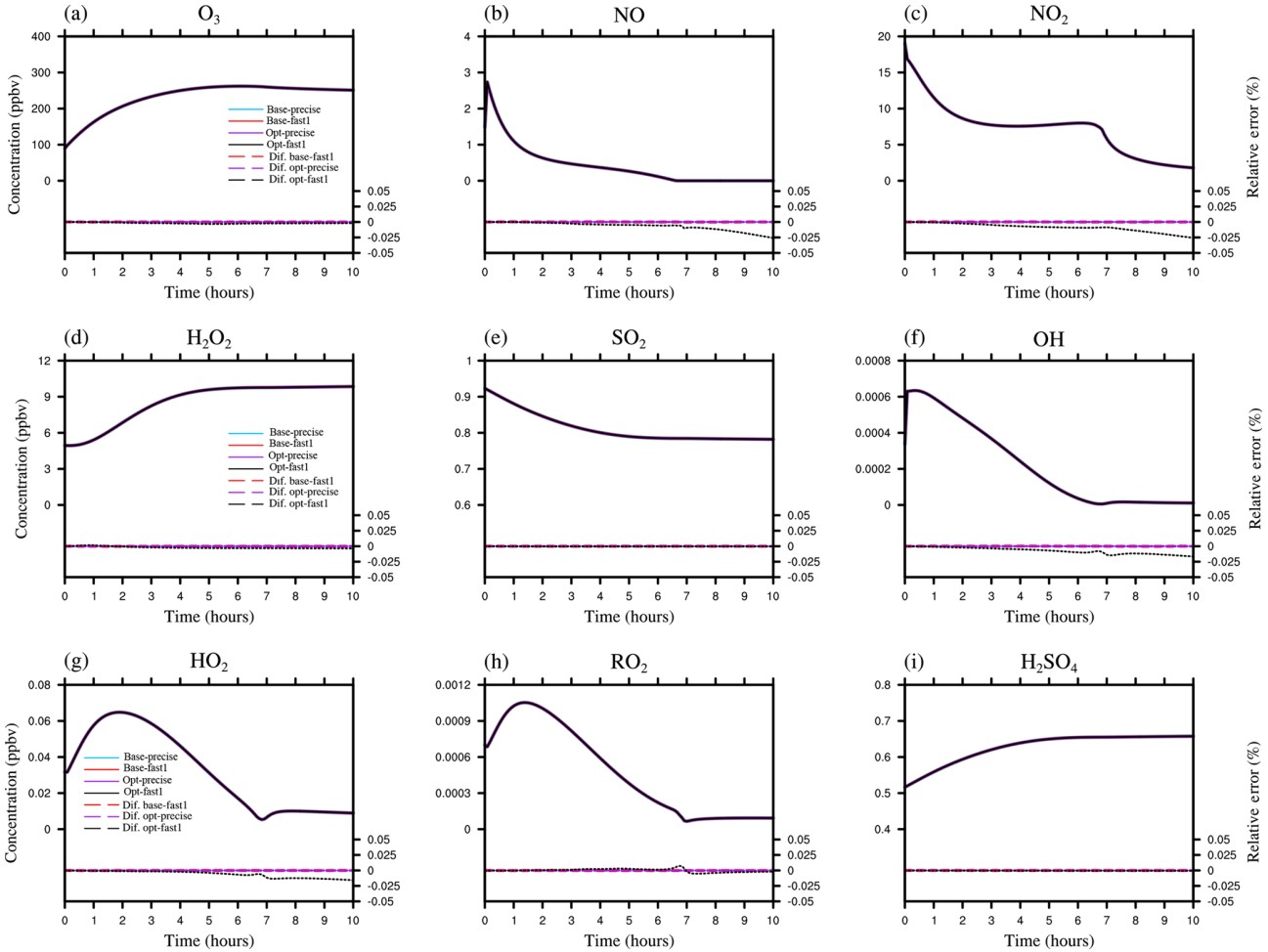

**Figure 4.** Comparison of the time-series concentrations of $O_3$, NO, $NO_2$, $H_2O_2$, $SO_2$, OH, $HO_2$, $RO_2$, and $H_2SO_4$ **(a–i)** from the baseline and optimized CBM-Z simulation with diverse *-fp-model* options. The simulation results by the baseline code with the *-fp-model precise* compile flag was as the benchmark. The solid lines show the time-series concentrations of the species from different experiments and the dashed lines showed the relative errors (RE) of simulated concentrations between the benchmark and the results by other combinations of the code and *-fp-model* options.

**Table 4.** The performance tests of the baseline and optimized code on different CPUs and KNL platforms with one physical cores. The unit of the wall times for the tests is seconds (s).

|               | Processor        | Vector instruction | *-fp-model*      | Wall time         | Speedup        |
|---------------|------------------|--------------------|------------------|-------------------|----------------|
| Baseline CBM-Z | Xeon E5-2680 V4 | AVX2               | precise<br>fast=1 | 1014.67<br>792.03 | 1.00<br>1.28   |
|               | Xeon Gold 6132   | AVX512             | precise<br>fast=1 | 665.44<br>497.64  | 1.52<br>2.04   |
| MP CBM-Z      | Xeon E5-2680 V4  | AVX2               | precise<br>fast=1 | 581.14<br>153.32  | 1.75<br>6.62   |
|               | Xeon Gold 6132   | AVX512             | precise<br>fast=1 | 352.00<br>55.42   | 2.88<br>18.31  |
|               | Xeon Phi 7250    | AVX512             | precise<br>fast=1 | 3454.90<br>214.09 | 0.29<br>4.74   |

and can be important in making full use of the new MIC architecture processors such as KNL.

### 3.3 CTM test

The regional CTM, the NAQPMS (Wang et al., 2001; ZiFa et al., 2006), was used to test the MP CBM-Z module under more realistic conditions. The following subsections will describe the CTM test case and will present results from the scientific validation and its computational performance.

#### 3.3.1 CTM test case description

The NAQPMS is a regional CTM developed by IAP, CAS (Li et al., 2011, 2013), and has been widely used in air quality research (Wang et al., 2018) and routine air quality forecasting (Wu et al., 2010; Chen et al., 2013). NAQPMS involves all essential processes including diffusion, advection, dry and wet deposition, and multiphase chemistry reactions. More details can be found in Li et al. (2013). In a similar way to the box model test case, the NAQPMS with the baseline and optimized CBM-Z modules were compiled with various compile flags as shown in Table 3.

The test case is a 72 h simulation covering the East Asia region. The horizontal resolution is 15 km with $339 \times 432$ grid boxes. The model adopted 20 vertical layers. The meteorological fields driving the NAQPMS were provided by the Weather Research and Forecasting (WRF) model (Skamarock et al., 2008). The anthropogenic emission inventory was from the Hemispheric Transport of Air Pollution (HTAP) V2 and the biogenic emission inventory was provided by results from Sindelarova et al. (2014) using the Model of Emissions of Gases and Aerosols from Nature (MEGAN) (Guenther et al., 2006, 2012). The simulation started at 00:00 UTC, 17 August 2015, and ended at 00:00 UTC, 20 August 2015. We only used one node for testing to exclude the interference of network communication. Each experiment was repeated five times and the performance was assessed on the basis of the average value.

#### 3.3.2 CTM validation

We chose four major gas pollutants, i.e., $NO_2$, $O_3$, $SO_2$, and CO, after 72 h integration to evaluate the optimized code. The simulation results of the baseline NAQPMS code compiled by the *precise* flag were as the benchmark results, and we mainly compared the simulation results of the baseline NAQPMS code with the *fast=1* flag and the optimized NAQPMS with *precise* and *fast=1*.

Figures 5 and 6 present the spatial distributions of $NO_2$, $O_3$, $SO_2$, and CO as well as the absolute errors (AEs) of their concentrations from other experiments relative to the baseline. We find that all model results show the same spatial distribution of pollutants. In general, for $NO_2$, $O_3$, and $SO_2$, the AEs in the majority of grid boxes are in the range of $\pm 0.02$ ppbv for the three experiments; for CO, the AEs of

baseline and optimized NAQPMS with the same *fast* = 1 are outside that range, showing more obvious AEs than that of other species.

The *precise* option enables the results of the two versions to be more consistent. Figure 7 shows the distribution of AEs and relative error (REs) for four species in the near-surface model layer. For the majority of points, the AEs and REs are in a relatively small range. However, some points show exceptional and obvious errors. The maximum AEs for $NO_2$, $O_3$, $SO_2$, and CO are 0.166, 0.197, 0.001, and 0.03 ppb over the whole map after 72 h of integration, and the *fast=1* option shows more obvious errors for both versions. For the baseline NAQPMS code, using *fast=1* leads to maximum AEs of 0.23, 4.5, 0.17, and 2.6 ppbv for $NO_2$, $O_3$, $SO_2$, and CO, respectively. To NAQPMS with the MP CBM-Z, using the *fast=1* option leads to maximum 0.13, 0.93, 0.76, and 0.64 ppbv AEs for $NO_2$, $O_3$, $SO_2$, and CO over the whole domain, which is better than the baseline NAQPMS.

In addition to considering the accuracy mentioned above, the impact of the *-fp-model* option on performance should be considered. In some pragmatic applications like routine air quality prediction, it is reasonable to sacrifice accuracy to gain computational performance. Conversely, applications like long-term climate simulations, choosing safer compile flags, or adopting double-precision for calculations to avoid accumulation of errors.

#### 3.3.3 CTM computational performance

The performance of the baseline NAQPMS with *precise* was the benchmark for comparison with other tests. As shown in Table 6, in the original version of NAQPMS, the CBM-Z module accounts for 72.26 % of the wall-clock time for the whole simulation. Changing the compile option of *-fp-model* to improve performance by sacrificing accuracy leads to $1.34\times$ and $1.25\times$ speedups for the module CBM-Z and the whole model on the Intel Xeon E5-2680 platform, respectively. By updating the CPU from Intel Xeon E5-2680 to Intel Xeon Gold 6132, the module CBM-Z and whole model gain $1.28\times$ and $1.29\times$ speedups, respectively. The speedups improve to $1.68\times$ and $1.58\times$ for CBM-Z and the whole model, respectively, by using the *fast=1* compile flag on the Xeon Gold 6132 platform. The benefit from updating hardware is limited with the baseline code and supports the need for optimizing code to adapt to the new hardware features.

The computational performance of the gas-phase chemistry module and the NAQPMS are largely improved after adopting the MP CBM-Z, as described in this paper. As shown in Table 6, the CBM-Z model and the whole NAQPMS shows speedups of $1.59\times$ and $1.40\times$ on the old Xeon E5-2680 platform with the same *precise* compile flag, and the speedups are improved to $4.45\times$ and $2.44\times$ by using the *fast=1* compile flag. With the same *fast=1* flag, the MP CBM-Z showed 3.32 and 1.96 times acceleration compared with the baseline CBM-Z for the gas-phase chemistry

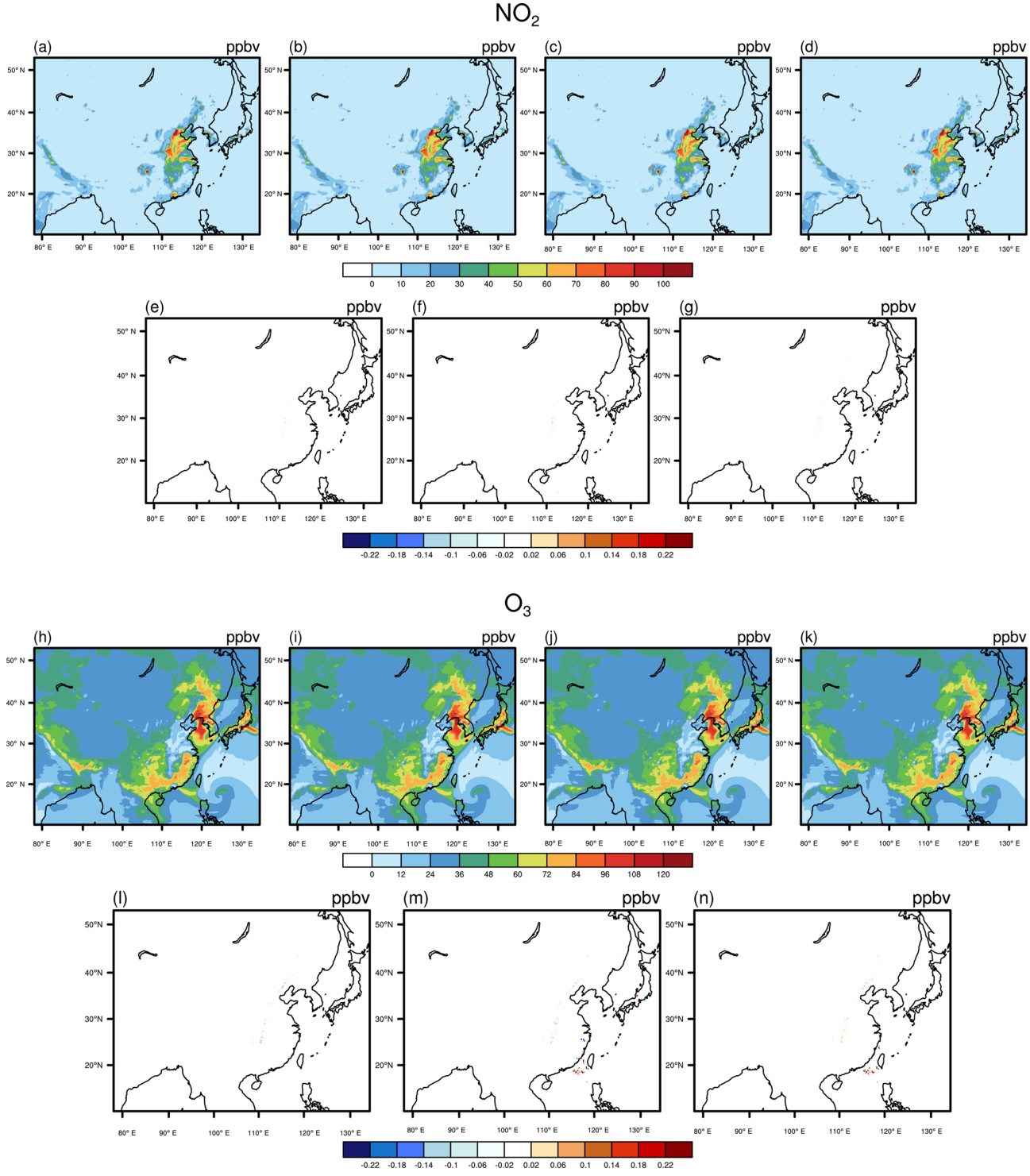

**Figure 5.** $NO_2$ and $O_3$ concentrations outputted by baseline and optimized code with different accuracy compile flags. Panels **(a)** and **(h)** are from baselines code compiled by the *precise* option, which are treated as benchmark for comparison. Panels **(b)** and **(i)** are from optimized code compiled by the *precise* option. Panels **(c)** and **(j)** are from baseline codes compiled by the *fast=1* flag. Panels **(d)** and **(k)** are from optimized code compiled by the *fast=1* flag. Panels **(e–g)** and **(l–m)** are the output concentration differences of optimized code (precise), baseline code (fast=1), and optimized code (fast=1) compared with baseline code (precise).

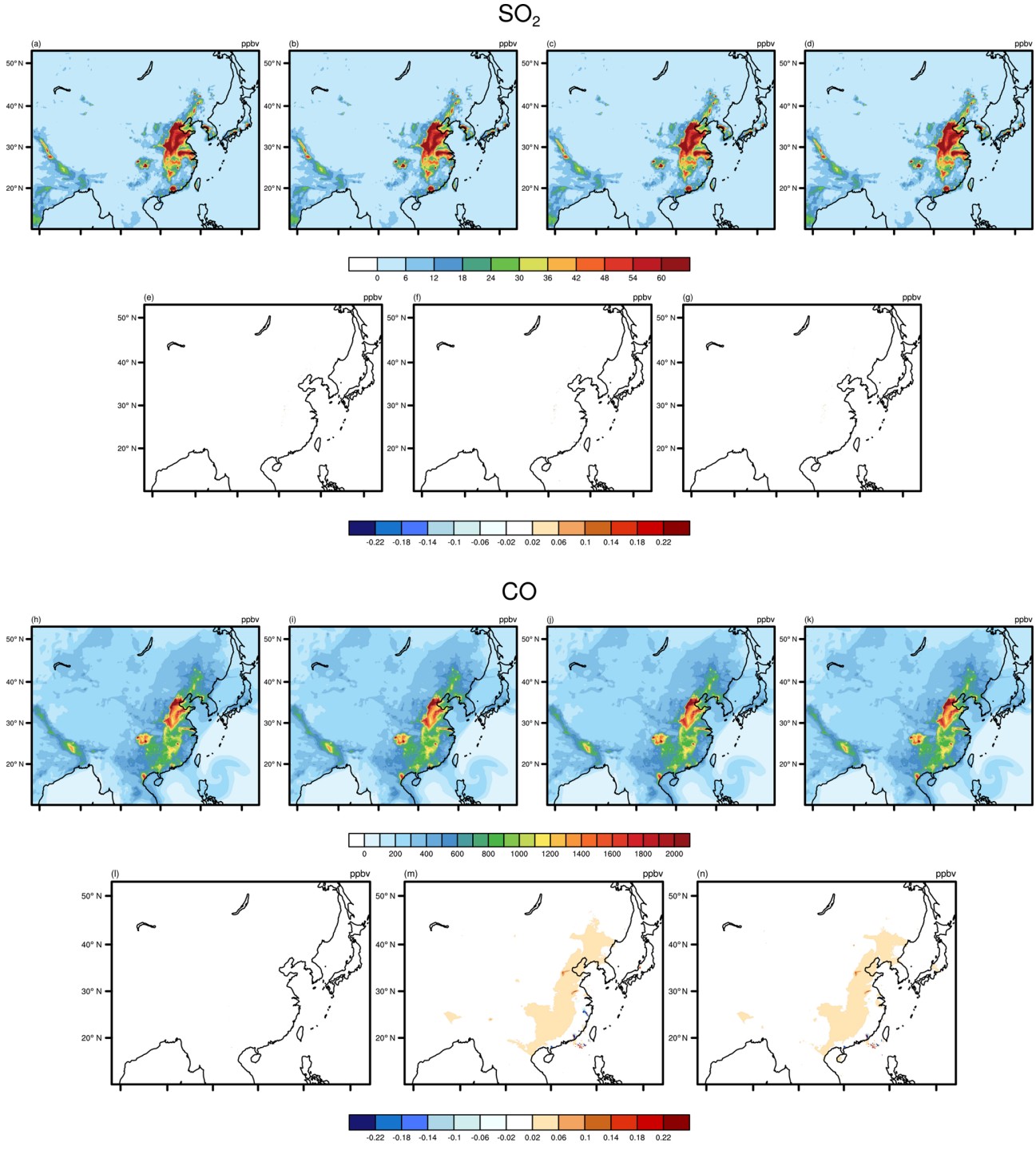

**Figure 6.** SO$_2$ and CO concentrations outputted by baseline and optimized code with different accuracy compile flags. Panels **(a)** and **(h)** are from baselines code compiled by the *precise* option, which are treated as benchmark for comparison. Panels **(b)** and **(i)** are from optimized code compiled by the *precise* option. Panels **(c)** and **(j)** are from baseline codes compiled by the *fast=1* flag. Panels **(d)** and **(k)** are from optimized code compiled by the *fast=1* flag. Panels **(e–g)** and **(l–m)** are the output concentration differences of optimized code (precise), baseline code (fast=1), and optimized code (fast=1) compared with baseline code (precise).

**Table 5.** The performance tests of the optimized code on different CPUs and KNL platforms with MPI and OpenMP. The unit of the wall times for the tests is seconds (s).

|  | Single core test | | | | |
|---|---|---|---|---|---|
|  | Processor | Vector instruction | Number of cores | Wall time | Speedup |
| MP CBM-Z | Xeon E5-2680 V4 | AVX2 | 1 | 792.03 | 1.00 |
|  | MPI with vectorization | | | | |
|  | Xeon E5-2680 V4 | AVX2 | 28 | 7.57 | 104.63 |
|  | Xeon Gold 6132 | AVX512 | 28 | 3.99 | 198.50 |
|  | Xeon Phi 7250 | AVX512 | 68 | 4.52 | 175.23 |
|  | OpenMP with vectorization | | | | |
|  | Xeon E5-2680 V4 | AVX2 | 28 | 7.84 | 101.02 |
|  | Xeon Gold 6132 | AVX512 | 28 | 4.07 | 194.60 |
|  | Xeon Phi 7250 | AVX512 | 68 | 4.73 | 167.45 |

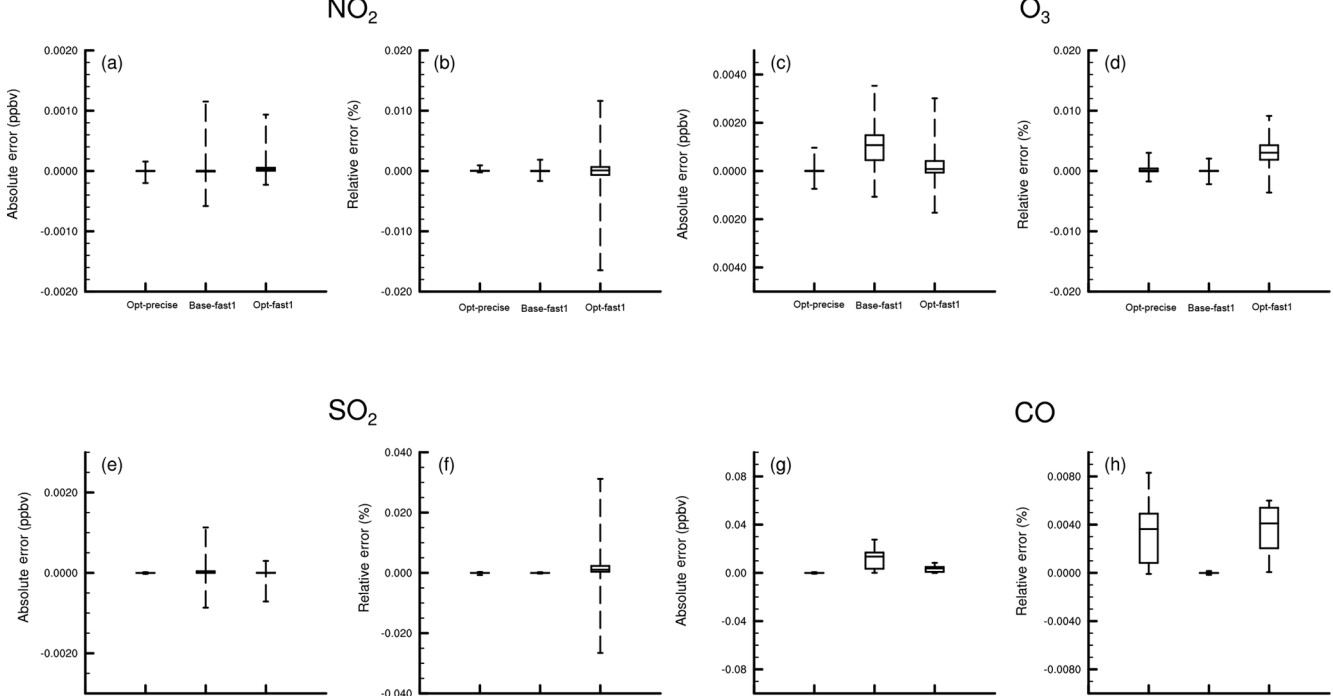

**Figure 7.** The distributions of absolute errors and relative errors for $O_3$, $NO_2$, $SO_2$, and CO in the near-surface model layer. The reference points are 1 %, 25 %, 50 %, 75 %, and 99 %.

module and whole NAQPMS. Such results illustrate that the optimization for vectorization improves the potential on existing hardware and the performance is highly improved even with the relatively strict *precise* compile flag, which prevents most vectorizations.

The new generation CPU further improves the performance of the MP CBM-Z. Using the platform with the new generation processor Xeon Gold 6132, the speedups reach $1.84\times$ and $1.66\times$ for the CBM-Z and the NAQPMS with the *precise* compile flag, respectively, and adopting the *fast=1* compile flag improves the speedups to $8.22\times$ and $3.50\times$ compared with the benchmark performance. On the same Xeon Gold 6132 platform with the *fast=1* compile flag, the MP CBM-Z gains 3.32 and 2.22 times acceleration compared with the baseline CBM-Z for the gas-phase chemistry module and the whole NAQPMS. Moreover, the proportion of time taken by the gas-phase chemistry declined to 30.74 % compared to 72.26 % in the baseline model.

**Table 6.** The performance tests of the baseline and optimized code on the diverse platforms with different compile flags. The unit of the wall times for the tests is seconds (s).

| | Vector processor | Instruction | Wall time *-fp-model* | Wall time (CBMZ) | Speedup (Total) | Speedup (CBMZ) | (Total) |
|---|---|---|---|---|---|---|---|
| Baseline NAQPMS | Xeon E5-2680 V4 | AVX2 | precise | 17 675.86 | 24 460.54 | 1.00 | 1.00 |
| | | | fast=1 | 13 201.56 | 19 619.20 | 1.34 | 1.25 |
| | Xeon Gold 6132 | AVX512 | precise | 13 817.24 | 18 950.95 | 1.28 | 1.29 |
| | | | fast=1 | 10 544.60 | 15 502.39 | 1.68 | 1.58 |
| NAQPMS with MP CBM-Z | Xeon E5-2680 V4 | AVX2 | precise | 11 127.90 | 17 454.95 | 1.59 | 1.40 |
| | | | fast=1 | 3971.48 | 10 019.21 | 4.45 | 2.44 |
| | Xeon Gold 6132 | AVX512 | precise | 9584.59 | 14 698.38 | 1.84 | 1.66 |
| | | | fast=1 | 2150.20 | 6994.43 | 8.22 | 3.50 |
| | Xeon Phi 7250 | AVX512 | fast=1 | 2997.96 | 19 239.20 | 5.90 | 1.27 |

In addition, the MP CBM-Z extends the benefit gained from advanced hardware. Using the same fast=1 compile option, the performance of the baseline CBM-Z on the AVX-512 platform is about 1.25 times that CE2 on the AVX-2 platform, and the performance of the MP CBM-Z is about 1.84 times of that on AVX-2 platform. The efficiency of using the new CPUs improved by about 47 % by adopting the MP CBM-Z. Therefore, enhancing the vectorization of code ensures that applications, like the CTM in this paper, could further utilize the improvement of processors on vectorization in the future.

KNL are more reliant on SIMD for performance according to the test results. The CBM-Z module is accelerated on KNL with a speedup of 5.9×, but the whole model only achieved a 1.27 times acceleration compared with the benchmark performance. Comparing the baseline CBM-Z on the Intel Xeon Gold 6132 platform, the MP CBM-Z achieves a speedup of 3.52× for the gas-phase chemistry on KNL; however, the performance of the whole model declined by 24 %. Therefore, the MP CBM-Z largely improved the efficiency of CBM-Z on KNL by improving its vectorization, but further optimizations are required for greater efficiency of the whole CTM on the KNL architecture.

## 4 Conclusions and discussion

A new framework was designed for helping the chemical kinetics kernel CBM-Z to adapt to the next-generation processes by improving its vectorization. Through packing multiple spatial points, the optimized CBM-Z module handled these simultaneously. The functions in the original CBM-Z were restructured with loops, which provided the opportunity to implement the fine-grain level parallelization of vectorization. Meanwhile, we masked the heterogeneous grid boxes to integrate the chemistry sub-schemes in the CBM-Z to perform the calculation of multiple grid boxes simultaneously.

Since the contiguous grid boxes have similar chemistry processes, the impact of this on the scientific performance was largely limited, and the code was highly vectorized.

The computation cluster equipped with two generation CPUs (Intel Xeon E5-2680 V4 and Intel Xeon Gold 6132) and KNL (Intel Xeon Phi 7250) provided by IAP, CAS, were used to test the performance. We tested the code with two different compile options of *-fp-model precise* and *-fp-model fast=1* to present its impact on the accuracy of single-precision computation and performance. The validation test ensured the reliability of our optimization on the model results, and the errors in all diagnostic chemical species caused by the single float calculations were lower than about 0.025 % after 10 h integration with the *fast=1* option. Based on the HPC performance characteristic from the Intel VTune tools on the Intel Xeon Gold 6132, the GFLOPS of CBM-Z increased from 4.81 to 21.37, and the vector capacity usage improved from 14.30 % in the baseline CBM-Z to 89.40 % in the optimized CBM-Z.

The tests using the single core showed that the vectorization optimization led to speedups of 5.16× and 8.97× on Intel Xeon E5-2680 V4 and Intel Xeon Gold 6132 CPUs, respectively, and KNL achieves a speedup of 3.69× compared with the baseline CBM-Z on the Intel Xeon E5-2680 V4 platform. It highlights the importance of vectorization on the KNL platform. Meanwhile, we also tested the MPI and OpenMP version of CBM-Z. The speedup on the two generation CPUs can reach 104.63× and 198.50× using MPI and 101.02× and 194.60× using OpenMP, respectively, and the speedup on the KNL node can reach 194.60× using MPI and 167.45× using OpenMP. The speedup on the KNL node can reach 194.60× using MPI and 167.45× using OpenMP. The speedup of the optimized CBM-Z is approximately 40 % higher on a one-socket KNL platform than on a two-socket Broadwell platform and about 13 %–16 % lower than on a two-socket Skylake platform.

The regional CTM NAQPMS was also used to test the practical improvement of the MP CBM-Z in more realistic scenarios. The baseline and optimized code of NAQPMS compiled with the *precise* and *fast=1* options, respectively, were tested on diverse platforms. The model outputs after 72 h simulation were used to evaluate the error by the code as well as the compile flags. The difference between the baseline and optimized code are generally in the range of $\pm 0.02$ ppbv using *precise*. The maximum discrepancy over the whole map is about 0.166, 0.197, 0.001, and 0.03 ppbv for $NO_2$, $O_3$, $SO_2$, and CO. The *fast=1* option leads to larger errors; however, computational performance could benefit a lot through adopting this option.

The results of the CTM test with the *fast=1* option show that the MP CBM-Z leads to a speedup of 3.32 and 1.96 for the gas-phase chemistry module and the CTM on the Intel Xeon E5-2680 platform, respectively. Moreover, on the new Intel Xeon Gold 6132 platform, the MP CBM-Z gains $4.90\times$ and $2.22\times$ speedups for the gas-phase chemistry module and the whole CTM. For the KNL, the MP CBM-Z enables a $3.52\times$ speedup for the gas-phase chemistry module, but the whole model lost 24.10 % performance compared to the CPU platform due to the poor performance of other modules. Since this optimization seeks to improve the utilization of the VPU, the model is more suitable for the new generation processors adopting the more advanced SIMD technology. The results of our tests already show that the benefit of updating CPU improved by about 47 % by using the MP CBM-Z since the optimized code has better adaptability for the new hardware.

In general, the choice of *-fp-model* compile flag decides the balance between accuracy and performance. According to our test, after using the *fast=1* option, the performance of the code is largely improved by sacrificing some accuracy. However, the loss of accuracy is relatively small, and in some practical applications that do not require high-accuracy floating-point calculations, it is acceptable to use the *fast=1* option.

Besides the CBM-Z chemical scheme, this algorithm is also suitable for models with a similar code structure to improve its vectorization. In addition, in this study, CBM-Z was treated as an example to describe this simple optimization strategy to implement the optimization on new generation processors, which emphasize the importance of vectorization. However, some specific strategies should also be considered before adoption. The optimizing methods such as constructing loops from the discrete scalar calculations as described in Wang et al. (2017), would diminish the readability of the source code by using indirect indexing and could cause problems to subsequent developers. Therefore, it is essential to adopt good practice, e.g., commenting code well and controlling the compile process, for ease of maintenance and development.

*Code availability.* The source code of the baseline and optimized version CBM-Z box model, including OpenMP and MPI versions, is available online via ZENODO (https://doi.org/10.5281/zenodo.1161576; Wang et al., 2018).

*Supplement.* The supplement related to this article is available online at: https://doi.org/10.5194/gmd-12-1-2019-supplement.

*Author contributions.* QW, HSC, XT, and ZFW planned and organized the project. JL and HW designed fine-grained parallelization algorithm for CBM-Z module, and QW HSC, XT, and XC took part in the discussion. JL, HW, HSC, and XT prepared the CBM-Z test cases and input datasets, and HW, HSC, and QW finished box model validation. XT, HSC, and ZWF prepared the NAQPMS code and its input dataset, HW coupled the MP CBM-Z module to NAQPMS, and QW, HW, and XT validated and discussed the model results. HW and JL analyzed the model performance data, and QW, HQC, and LW validated its reasonability. HW and QW wrote the manuscript. HW, JL, QW, XT, HSC, and XC revised the manuscript. HSC, XT, ZFW, XC, HQC, and LW reviewed and provided key comments on the paper.

*Competing interests.* The authors declare that they have no conflict of interest.

*Acknowledgements.* The National Key R&D Program of China (2017YFC0209805 and 2016YFB0200800), the CAS Information Technology Program (XXH13506-302), the National Natural Science Foundation of China (41305121), and the Fundamental Research Funds for the Central Universities funded this work. The authors would like to thank the Institute of Atmospheric Physics and Intel Corporation's Software Support Group (SSG) for providing the high-performance computing (HPC) environment and technical support. The authors thank the topic editor and three anonymous referees for their valuable comments.

Edited by: Fiona O'Connor
Reviewed by: three anonymous referees

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

**Remarks from the language copy-editor**

CE1   Thank you for clarifying this. We would not usually use bold and italic together in this context. Would it be acceptable to make all instances italic only? I have done this for now, so you can see what it looks like and have also used italic font for function names. Please let me know if any instances have been missed - thank you!

CE2   Inserting "of" here would not be grammatically correct.

**Remarks from the typesetter**

TS1   We are not sure about this change. This needs the confirmation by the editor. Please write a sentence for the editor explaining why this should be changed. Thank you.