# Peer review of "MP CBM-Z V1.0: design for a new CBM-Z gas-phase chemical mechanism architecture for next generation processors"

_Geoscientific Model Development, 2018_

## Referee Comment (RC1) · Anonymous Referee #1 · 15 Jun 2018

In this manuscript, the authors design a architecture for CBM-Z chemical mechnism on next generation processors. This is interesting and quite useful for the routine numerical air quality forecast. I believe that this accelration is helpful to policy managers. The accelration of chemical solver is a difficult problem since 1980s. The computer technique has a rapid development. However, air quality models do not fully utilize this development. The contribution is generally well-written and complete. I suggested this manuscript to be published after considering the following comments.

Specific Comments: 1. In figure 4, The authors plots the intercomparison of SO2. O3, H2O2, NO, H2SO4 between base and optimized simulations. I suggested some

short-lived species like OH, HO2, RO2 should be compared, becaused these species is more sensitive to the mechinism, and very important to atmospheric oxdiation. 2. The scenario in this manuscript is urban/polluted conditions. The authors presents comparisons in other scenario like marine, biomass burning.

---

## Short Comment (SC1) · 19 Jun 2018

**General comment**

In this manuscript, the authors design an architecture for CBM-Z chemical mechanism on next generation processors. This is interesting and quite useful for the routine numerical air quality forecast. I believe that this acceleration is helpful to policy managers. The acceleration of chemical solver is a difficult problem since 1980s. The computer technique has a rapid development. However, air quality models do not fully utilize this development. The contribution is generally well-written and complete. I suggested this manuscript to be published after considering the following comments.

Response: The authors thank for your time and your encourage for this manuscript. As the reviewer mentioned, the hardware of computer has developed and improved, but the air quality model has not fully taken advantage of it to improve its computation efficiency, which may limit its practical value. This work is a part of the project about improving the computation performance of Global Nested Air Quality Prediction Model System (GNAQPMS) (Chen et al., 2015;Wang et al., 2017) and exploring the potential of the Intel Many Integrated Core (MIC) Architecture Chips on air quality model. We hope this work could be a good example to help other users to improve the calculation efficiency of their models on the new generation chips.    The detailed responses to the specificcomments are given point to point as following.

**Specific Comments**

1. In figure 4, The authors plot the intercomparison of SO2. O3, H2O2, NO, H2SO4 between base and optimized simulations. I suggested some short-lived species like OH, HO2, RO2 should be compared, because these species is more sensitive to the mechinism, and very important to atmospheric oxdiation.

Response: The authors appreciate your constructive comments. We fully agree with your suggestion and will add the comparison of short-lived species like OH, HO2, RO2 in the revised manuscript to further verify the model results.

2. The scenario in this manuscript is urban/polluted conditions. The authors present comparisons in other scenario like marine, biomass burning.

Response: The authors appreciate your important comments and accept your advice. Our following work is to conduct more experiments to test the efficiency as well as the validity of the new codes, including the scenario mentioned by the reviewer. We will present corresponding results in the revised manuscript.

**Reference**

Chen, H. S., Wang, Z. F., Li, J., Tang, X., Ge, B. Z., Wu, X. L., Wild, O., and Carmichael, G. R.: GNAQPMS v1.0, a global nested atmospheric mercury transport model: model description, evaluation and application to trans-boundary transport of Chinese anthropogenic emissions, Geosci. Model Dev., 8, 2857-2876, 10.5194/gmd-8-2857-2015, 2015.

Wang, H., Chen, H., Wu, Q., Lin, J., Chen, X., Xie, X., Wang, R., Tang, X., and Wang, Z.: GNAQPMS v1.1: accelerating the Global Nested Air Quality Prediction Modeling System (GNAQPMS) on Intel Xeon Phi processors, Geosci. Model Dev., 10, 2891-2904, 10.5194/gmd-10-2891-2017, 2017.

---

## Referee Comment (RC2) · Anonymous Referee #2 · 22 Jun 2018

This paper describes improvements in vectorisation of the Carbon Bond Mechanism Z. While interesting, significant problems need to be addressed before I can recommend its publication in GMD.

**General Comments**

1. The manuscript contains numerous grammatical and spelling mistakes, and I will attempt to highlight these in detail in my specific comments below. I would recommend the authors check any resubmission carefully.

2. Throughout the manuscript the authors make reference to "the chemistry transport model", "the CTM", and "the air quality model" without first describing which specific

CTM or AQ model they are referring to, if any. I believe from the context that the model the authors use is in fact the GNAQPMS model, first mentioned on P2L13. If the authors mean a specific CTM it should be discussed in this context, although in some places (e.g. P3L15) the authors mean CTMs in general. If the authors mean any CTM then the phrase "Chemistry Transport Models" or "CTMs" would be appropriate.

3. The paper itself only covers improvements to the CBM-Z module, which I assume is included in some way into GNAQPMS, although this is not discussed by the authors. The CBM-Z code is provided by the authors on Zenodo, which is great to see, although I find it difficult to understand the improvements made in the code by examining Figures 2 and 3 in the manuscript. While graphical representations of these optimisations are useful, I would like to see how these were implemented in practice by the authors giving specific code examples within the manuscript.

4. The authors only give results from a CBM-Z standalone model, rather than having incorporated these improvements back into their CTM and presenting results from there. If available, I would certainly like to see what this does to model performance, as I feel it would strengthen this work greatly. As it is, they present only two case studies, where emissions are zero and meteorological conditions were constant. I would expect to see simulations of a number of different environments similar to those seen in simulations, i.e. free troposphere, boundary layer, urban, rural etc.

5. CBM-Z output is plotted for 10 model hours, and within this time the relative error introduced by the optimisations is less than 0.05

6. The authors describe running the CBM-Z model using a single point and over a number of grids for testing, which I believe to be a spatial grid from the context. In this case, is there any transport or mixing between grid-points, or any differences in meteorological variables? If not, how was this configured and set-up - is each point solving the same conditions at all times? If so, this will not be representative of real-world usage where there will be a large amount of variation across the domain. For the

single-point model, is it integrated in time? If not, how is it configured? I again question whether it would be better to perform these 3D simulations with a CTM instead of a stand-alone CBM-Z model, as it would then allow for a better understanding of how these improvements impact their model in practice. If this is not possible, the authors should give reasons as to why this is.

My major concerns with the manuscript as presented are:

A. Insufficient conditions used for testing

B. Insufficient analysis of the errors introduced by the optimisations.

**Specific Comments and Technical Corrections**

P1L11 - computational

P1L12 - model used.

P1L18 - Knights Landing

P1L19 - I question whether the <0.05

P1L29 - Chemistry Transport Models

P2L1 - a CTM

P2L1 - computational

P2L3 - relatively simple processes are adopted in CTMs to minimize

P2L4 - computational

P2L5-6 - I would suggest this: "Therefore, air quality simulation studies can benefit significantly by improving the performance of the CTM used."

P2L16 - what is meant by "improving the frequency of air quality forecasting" in this context?

P2L19-21 - do you have any references for the trend in changes to computing architecture that can be quoted here?

P3L4 - what is "the air quality model" in this context? Do you mean "in several air quality models"?

P3L9 - what do you mean by "architecture" in this context?

P3L14 - tropospheric

P3L15 - in CTMs

P4L1 - delete "The" and just start the sentence "CBM-Z also..."

P4L6 - simulations

P4L14 - space after AVX-512

P4L19 - do you mean "implement fine-grained parallelization"?

P4L21 - fine-grained

P4L21 - delete the comma after SIMD

P5L5 - does the solver use a fixed number of iterations, or does it integrate to convergence?

P5L13-14 - does the fact that calculations are performed on all grids but not all grids are copied back introduce a possible inefficiency? Are these grids taking time to solve that could be better spent doing something else?

P5L22-23 - as I have mentioned in the General Comments, how representative are these examples? How exactly were they configured?

P6L17 - as I have mentioned in the General Comments, how robust is the error of range <0.05

P7L4-9 - Please provide more details of these 3D simulations, as I am unclear exactly

how they were set-up.

P8L10 - It's great to see that the code is provided, but I found the structure provided confusing and the README provided lacking. However, this isn't part of this manuscript, but I would urge the authors to improve the documentation provided, include a directory listing with what the files do etc.

P10 - Caption to Table 2: versions

P11 - Table 3: How many times were these tests run? Is it possible to provide an error estimate for these numbers?

P12 - Caption to Figure 2: vectors

P12 - Figures 2: As mentioned in the General Comments, I believe that this manuscript would benefit from seeing how the code is altered with these optimizations.

P12 - Caption to Figure 3: grids

P13 - Figure 4: As mentioned in the General Comments, do these trends continue? How long until they become significant?

P13/P14 - Figures 4 5: Which simulations are these figures from, the single-point case or the 3D simulation? I would suggest more species are analyzed, covering a range of chemical lifetimes.
* * *

---

## Referee Comment (RC3) · Anonymous Referee #2 · 22 Jun 2018

My previous review had some formatting issues where 3 of the lines were truncated, and I'm sorry for any confusion caused. I include these comments below.

**General Comments**

5. CBM-Z output is plotted for 10 model hours, and within this time the relative error introduced by the optimisations is less than 0.05% (just), as seen in Figure 4 and mentioned throughout the manuscript. However, it is clear that this error is increasing for some species (e.g. H2O2, SO2, and H2SO4 as presented, and possibly others not shown). If these simulations were run for longer than 10 hours, would these errors still be below 0.05%? For confidence in the improvements described I would expect to see

that the errors remain low for the length of a typical CTM simulation, which could be years depending on how the author's CTM is used. If the conditions used are more realistic (see point 4 above) do these errors increase?

**Specific Comments and Technical Corrections**

P1L19 - I question whether the <0.05% figure is correct (see General Comments)

P6L17 - as I have mentioned in the General Comments, how robust is the error of range <0.05%?

---

## Referee Comment (RC4) · Anonymous Referee #3 · 4 Jul 2018

This paper designed a new framework for an air quality model to adapt the new Intel architecture to improve the computing performance, which is very useful for HPC, model developer. I therefore fell that the manuscript is suitable for Geoscientific Model Development and Major Revision is needed. Some comments and suggestions follow below. I hope this paper can be more useful through further revisions.

1. Writing The writing is not good. I think the authors should look for a native language person to polish it. Some sentences are hard to understand, especially in 3.3 Performance Test and figure captions.

2. Experiments and analysis
a. The authors did not give the information about the floating precision of the model. Double precision, or single precision? The performance should be difference, I think. If possible, the authors had better show the results of model with both single and double precision.

b. In this paper, the authors only give the results with one node. How about the model's captions? Is the performance (speedup) on KNL still better than CPU?

c. In performance test, authors only test the MPI and OpenMP separately. It's very interesting that the performances are almost same. Can authors explain that? I just wonder how about the computing performance by using MPI/OpenMP hybrid parallel method?

d. It takes 539.86s with one CPU core while 4481.10s with one KNL core. The authors blame the worse performance to lower frequency of KNL. It's difficult to understand because frequency difference between CPU and KNL is significantly and much less than the computing difference. Other factors, such as memory bandwidth, should also contribute the computing difference.

———————————————————

---

## Author Comment (AC1) · 18 Aug 2018

General comments

This paper designed a new framework for an air quality model to adapt the new Intel architecture to improve the computing performance, which is very useful for HPC, model developer. I therefore fell that the manuscript is suitable for Geoscientific Model Development and Major Revision is needed. Some comments and suggestions follow below. I hope this paper can be more useful through further revisions.

Response: Thank you for your encourage and sharing the time in this manuscript. The

detailed responses to the comments are given point to point as followed.

Specific comments

1. Writing The writing is not good. I think the authors should look for a native language person to polish it. Some sentences are hard to understand, especially in 3.3 Performance Test and figure captions.

Response: The authors appreciate your precious comments. We would take some measures, including inviting a native speaker, to improve the language of the transcript, especially the part in section 3.3 mentioned by the reviewer.

2. Experiments and analysis a. The authors did not give the information about the floating precision of the model. Double precision, or single precision? The performance should be difference, I think. If possible, the authors had better show the results of model with both single and double precision. Response: Currently, the floating precision of the model is single-precision. That is a good idea. We will follow your advice and conduct the relevant experiments to test the difference of the performance with single and double precision.

b. In this paper, the authors only give the results with one node. How about the model's captions? Is the performance (speedup) on KNL still better than CPU?

Response: The authors appreciate your constructive comments. As the reviewer mentioned, we only conducted the single node tests currently. We will follow your advice to do some tests to investigate and compare the model's captions on CPU and KNL.

c. In performance test, authors only test the MPI and OpenMP separately. It's very interesting that the performances are almost same. Can authors explain that? I just wonder how about the computing performance by using MPI/OpenMP hybrid parallel method?

Response: The authors thank your for your precious comments. The MPI and OpenMP in our code were used for the same parallel segment in CBM-Z module, their acceler-

ation ratio is similar at the algorithm level. And we only took the tests on single node that there is no overhead of MPI on message passing cross-node through the network, which led to the similar performance of OpenMP and MPI. We will use the performance analyzing tools like Intel Trace Analyzer and Collector (ITAC) and Intel Vtune to further analyze the codes. The MPI and OpenMP are both used for the main loop currently in the CBM-Z module, and it requires more time and specific technological details for us to add MPI and OpenMP into different levels of parallel segments of CBM-Z module. In this study, we would mainly focus on the exiting codes and results, and the hybrid parallel method could be a direction for our future work.

d. It takes 539.86s with one CPU core while 4481.10s with one KNL core. The authors blame the worse performance to lower frequency of KNL. It's difficult to understand because frequency difference between CPU and KNL is significantly and much less than the computing difference. Other factors, such as memory bandwidth, should also contribute the computing difference.

Response: The authors thank for your comments. As the reviewer mentioned, other factors like memory bandwidth could also contribute to the gap of the performance. We will conduct some relevant experiments and use the tools like Intel Vtune and ITAC to investigate the potential reasons for the phenomenon.

Please also note the supplement to this comment:
https://www.geosci-model-dev-discuss.net/gmd-2018-42/gmd-2018-42-AC1-supplement.pdf

---

## Author Comment (AC2) · 18 Aug 2018

This paper describes improvements in vectorisation of the Carbon Bond Mechanism Z. While interesting, significant problems need to be addressed before I can recommend its publication in GMD.

Response: The authors thank you for encourage and sharing your time in this manuscript. We have merged your two files of comments and the detailed responses to the comments are given point to point as followed, and the modifications followed the comments will be added into the revised manuscript.

[Figure]

General Comments

1. The manuscript contains numerous grammatical and spelling mistakes, and I will attempt to highlight these in detail in my specific comments below. I would recommend the authors check any resubmission carefully.

Response: We have modified these grammatical and spelling mistakes of the current manuscript following your guides, and we will check the revised manuscript carefully. And a native speaker will be invited to help us to improve the language before the resubmission.

2. Throughout the manuscript the authors make reference to "the chemistry transport model", "the CTM", and "the air quality model" without first describing which specific CTM or AQ model they are referring to, if any. I believe from the context that the model the authors use is in fact the GNAQPMS model, first mentioned on P2L13. If the authors mean a specific CTM it should be discussed in this context, although in some places (e.g. P3L15) the authors mean CTMs in general. If the authors mean any CTM then the phrase "Chemistry Transport Models" or "CTMs" would be appropriate.

Response: The authors agree with your comments. We would follow your advice and use the name of the GNAQPMS model when we referring to the specific model, meanwhile, we would use the phrase "Chemistry Transport Models" or "CTMs" when referring to CTMs in general.

3. The paper itself only covers improvements to the CBM-Z module, which I assume is included in some way into GNAQPMS, although this is not discussed by the authors. The CBM-Z code is provided by the authors on Zenodo, which is great to see, although I find it difficult to understand the improvements made in the code by examining Figures 2 and 3 in the manuscript. While graphical representations of these optimisations are useful, I would like to see how these were implemented in practice by the authors giving specific code examples within the manuscript.

Response: The authors appreciate your comments. Since we would only focus on the CBM-Z module and isolate the impact of other modules in this manuscript, we didn't show the results of GNAQPMS model and only discussed CBM-Z module. We will follow your advice and add the sample code to illustrate the implement of codes.

4. The authors only give results from a CBM-Z standalone model, rather than having incorporated these improvements back into their CTM and presenting results from there. If available, I would certainly like to see what this does to model performance, as I feel it would strengthen this work greatly. As it is, they present only two case studies, where emissions are zero and meteorological conditions were constant. I would expect to see simulations of a number of different environments similar to those seen in simulations, i.e. free troposphere, boundary layer, urban, rural etc.

Response: The authors thank for your comments. We accept your advices and will conduct more simulation experiments by using the CTM incorporated with the optimized CBM-Z model. We will test the performance of model under diverse scenarios, and corresponding results will be provided in the revised manuscript.

5. CBM-Z output is plotted for 10 model hours, and within this time the relative error introduced by the optimisations is less than 0.05% (just), as seen in Figure 4 and mentioned throughout the manuscript. However, it is clear that this error is increasing for some species (e.g. H2O2, SO2, and H2SO4 as presented, and possibly others not shown). If these simulations were run for longer than 10 hours, would these errors still be below 0.05%? For confidence in the improvements described I would expect to see that the errors remain low for the length of a typical CTM simulation, which could be years depending on how the author's CTM is used. If the conditions used are more realistic (see point 4 above) do these errors increase?

Response: The authors appreciate your constructive comments. We agree with your comments. Indeed, the error of some species is increasing with time, and we also concern about this issue. Therefore, we would further investigate the source of the

error by conducting more tests and try to find some solutions like using more reliable compiling flags to constrain the error. We will use the long-time simulation to see the trend of this error, and the CTM's simulation would be used to investigate its impact in real situations. In addition, more species with diverse chemical properties will be analyzed and results will be presented in the revised manuscript.

6. The authors describe running the CBM-Z model using a single point and over a number of grids for testing, which I believe to be a spatial grid from the context. In this case, is there any transport or mixing between grid-points, or any differences in meteorological variables? If not, how was this configured and set-up is each point solving the same conditions at all times? If so, this will not be representative of real-world usage where there will be a large amount of variation across the domain. For the single-point model, is it integrated in time? If not, how is it configured? I again question whether it would be better to perform these 3D simulations with a CTM instead of a stand-alone CBM-Z model, as it would then allow for a better understanding of how these improvements impact their model in practice. If this is not possible, the authors should give reasons as to why this is.

Response: The authors appreciate your comments. The meteorological variables, e.g. temperature, are varied with grids in the multiple grid test. There is also no transport as well as diffusion but only gas-chemistry process in that test to isolate the impact of other processes. We agree that this test is insufficient to illustrate the improvement in real world, and we will test results by using the CTM to further present the improvement of our optimized scheme. In addition, the single-point model is integrated in time as a box model to show the impact of optimization on results, and in the revised manuscript, we will also present the results of CTM to supplementarily its effect on model results.

My major concerns with the manuscript as presented are:

A. Insufficient conditions used for testing

Response: We fully understand your concern and we would use the CTM with the

optimized CBM-Z scheme to test the performance and validate the output of the model.

B. Insufficient analysis of the errors introduced by the optimisations.

Response: In the current manuscript, we have stated that using -O0 compiling flag could diminish the difference of the results, which demonstrate there is no logical and artificial errors of the optimized codes, but we didn't further investigate the impact of optimizing codes on results and how to constrain the error by using some specific compiling flags. Therefore, we will try to investigate the source of error and potential way to constrain the errors.

Specific Comments and Technical Corrections

P1L11 - computational

Response: We have accepted your advice and modified in the revised manuscript as following:

"Precise and rapid air quality simulation and forecasting are limited by the computational performance of the air quality model used, and the gas-phase chemistry module is the most time-consuming function in the air quality model."

P1L12 - model used.

Response: The authors appreciate your comments. We have accepted your advice and modified in the revised manuscript as following:

"Precise and rapid air quality simulation and forecasting are limited by the computational performance of the air quality model used, and the gas-phase chemistry module is the most time-consuming function in the air quality model."

P1L18 - Knights Landing

Response: We have accepted your advice and modified in the revised manuscript as:

"The Intel Xeon E5-2697 V4 CPU and Intel Xeon Phi 7250 Knights Landing (KNL) are

used as the benchmark processors."

P1L19 - I question whether the <0.05

Response: The authors appreciate your comments. The criteria of 0.05% comes from the results of single point test, and as mentioned above, we would further investigate the source of error by conducting more simulation experiments.

P1L29 - Chemistry Transport Models

Response: The authors appreciate your comments. We have accepted your advice and modified in the revised manuscript as following:

"As a useful tool for air quality problems, Chemistry Transport Models (CTMs) are widely used in studies of air quality"

P2L1 - a CTM

Response: The authors appreciate your comments. We have accepted your advice and modified in the revised manuscript as following:

"As the core of the AQF system, a CTM requires a large number of computational resources to simulate the complex chemical and physical processes."

P2L1 - computational

Response: The authors appreciate your comments. We have accepted your advice and modified in the revised manuscript.

"As the core of the AQF system, a CTM requires a large number of computational resources to simulate the complex chemical and physical processes."

P2L3 - relatively simple processes are adopted in CTMs to minimize

Response: The authors appreciate your comments. We have accepted your advice and modified in the revised manuscript as:

"To satisfy the demand of routine air quality forecasting in a timely manner, coarse spatial resolution and relatively simple processes are adopted in CTMs to minimize the use of computational resources."

P2L4 - computational

Response: The authors appreciate your precious comments. We have accepted your advice and modified in the revised manuscript as following

"To satisfy the demand of routine air quality forecasting in a timely manner, coarse spatial resolution and relatively simple processes are adopted in CTMs to minimize the use of computational resources."

P2L5-6 - I would suggest this: "Therefore, air quality simulation studies can benefit significantly by improving the performance of the CTM used."

Response: The authors appreciate your constructive suggestion. We have accepted your advice and modified in the revised manuscript as following:

"Therefore, air quality simulation studies can benefit significantly by improving the performance of the CTM used."

P2L16 - what is meant by "improving the frequency of air quality forecasting" in this context?

Response: The authors appreciate your comments. Yes, it means "improving the frequency of air quality forecasting" in the context and we have revised this sentence as following:

"The AQF system can also benefit from the performance improvement by adopting a higher model resolution and improving the frequency of air quality forecasting."

P2L19-21 - do you have any references for the trend in changes to computing architecture that can be quoted here?

Response: The authors appreciate your comments. Yes, Xu et al. (2015) and Lawrence et al. (2018) could be the appropriate reference here.

P3L4 - what is "the air quality model" in this context? Do you mean "in several air quality models"?

Response: The authors appreciate your comments. The "air quality model" here refers to the "CTMs". Following the second general comments of the reviewer, we would use the identical phase "CTMs" in the revised manuscript.

P3L9 - what do you mean by "architecture" in this context?

Response: The authors appreciate your comments. The phase "structure" may be more appropriate in this place, and it means we modified the codes structure of CBM-Z scheme, as we describe in the context, to improve the vectorization of codes.

P3L14 – tropospheric

Response: We have accepted your advice and modified in the revised manuscript as:

"CBM-Z is a lumped-structure photochemical mechanism that was developed to meet the needs of city-scale to global-scale tropospheric chemical simulation"

P3L15 - in CTMs

Response: We have accepted your advice and modified in the revised manuscript as following:

"The original scheme contains 67 species and 132 reactions. CBM-Z has been widely used in CTMs"

P4L1 - delete "The" and just start the sentence "CBM-Z also. . ."

Response: The authors appreciate your comments. We have accepted your advice and revised in the manuscript as:

"CBM-Z module still contains many scalar operations."

P4L6 - simulations

Response: The authors appreciate your comments. We have accepted your advice and modified in the revised manuscript as:

"Fortunately, contiguous model grids may have similar chemical processes in air quality simulations"

P4L14 - space after AVX-512

Response: The authors appreciate your comments. We have accepted your advice and modified in the revised manuscript.

P4L19 - do you mean "implement fine-grained parallelization"?

Response: The authors appreciate your comments. Yes, it is exactly what we mean and we have modified this part.

P4L21 - fine-grained

Response: The authors appreciate your comments, and we have revised in the manuscript as:

"Our goal is to implement fine-grained parallelization based on the SIMD and the grids that are distributed to a specific processor operate in parallel using the VPUs on each core."

P4L21 - delete the comma after SIMD

Response: The authors appreciate your comments. We have accepted your advice and modified in the revised manuscript.

P5L5 - does the solver use a fixed number of iterations, or does it integrate to convergence?

Response: The authors appreciate your comments. The solver uses the explicit algorithm and it didn't need iterations. The solver in this test is Modified-Backward-Euler

(MBE) method. The detailed description of the solver could be found in Feng et al. (2015) and Feng et al. (2017).

P5L13-14 - does the fact that calculations are performed on all grids but not all grids are copied back introduce a possible inefficiency? Are these grids taking time to solve that could be better spent doing something else?

Response: The authors appreciate your comments. No, this process can keep the high computational efficiency of Vector Processing Union (VPU) considering that the chemistry processes among grids are different. It avoids the logical judgements to assort the grids so that corresponding computations can be finished on VPU simultaneously, which means that the computations of multiple grids could be finished at the same time without doing extra logical judgment processes.

P5L22-23 - as I have mentioned in the General Comments, how representative are these examples? How exactly were they configured?

Response: The authors appreciate your comments. As we mentioned in the manuscript, the goal of the single-point case is to validate the model's output for debugging logical or artificial errors, and the 3-dimension (3D) case is used for the performance testing. The 3D case is derived from the real CTM simulation, and the meteorological conditions like relative humility of this case is diverse among grids but constant during the simulation. The 3D case does not contain the emission, transport and diffusion process but only gas-chemistry process.

Following your advice, we would conduct some real-scenario simulation by using CTM. In addition, we will provide more clear descriptions of the test conditions as well as configuration in the revised manuscript.

P6L17 - as I have mentioned in the General Comments, how robust is the error of range <0.05

Response: The authors appreciate your comments. The criteria of 0.05% is based

**GMDD**

on our current test and we can't answer how robust of these criteria is without further test and investigation. So we will use the long-time simulation as well as the CTM's simulation to investigate the source and possible solutions for the errors.

P7L4-9 - Please provide more details of these 3D simulations, as I am unclear exactly how they were set-up.

Response: The authors appreciate your comments. The 3-dimension case is derived from the real CTM simulation, and the meteorological conditions like relative humility of this case is diverse among grids but constant during the simulation. The 3D case does not contain the emission, transport and diffusion process but only gas-chemistry process. We will provide more detail about the test cases in the revised manuscript

P8L10 - It's great to see that the code is provided, but I found the structure provided confusing and the README provided lacking. However, this isn't part of this manuscript, but I would urge the authors to improve the documentation provided, include a directory listing with what the files do etc.

Response: The authors appreciate your comments. We would follow your advice to provide the codes with detailed document as well as directory structure for readers' or users' convenient.

P10 - Caption to Table 2: versions

Response: The authors appreciate your comments, and we have revised caption to Table 2 in the manuscript as:

"Compile flags of the different versions of CBM-Z."

P11 - Table 3: How many times were these tests run? Is it possible to provide an error estimate for these numbers?

Response: The authors appreciate your comments. These tests were done only one time currently on the relative stable testing platforms. It is possible that some unpredictable situations would affect the results, but we will repeat the following tests more times to ensure the stability of results.

P12 - Caption to Figure 2: vectors

Response: The authors appreciate your comments, and we have accepted your advice in the revised manuscript as:

"The i and j loops, equaled latitude and longitude loops, were merged to construct one vector to reduce the number of unfilled vectors."

P12 - Figures 2: As mentioned in the General Comments, I believe that this manuscript would benefit from seeing how the code is altered with these optimizations.

Response: The authors appreciate your comments. We would provide the figures of sample codes to show the optimization processes.

P12 - Caption to Figure 3: grids

Response: The authors appreciate your comments and we have accepted your advice in the revised manuscript as:

"The flowchart shows the way to mask the heterogeneous girds to integrate grids to perform the vectorization operations according to the iregime values."

P13 - Figure 4: As mentioned in the General Comments, do these trends continue? How long until they become significant?

Response: The authors appreciate your comments. We are not sure about whether this error would continue. If these trends of error continue, it could become significant and effect the results. We also concern about this issue, therefore, we would further investigate the source of the error by conducting more tests and try to find some solution to figure it out.

P13/P14 - Figures 4 5: Which simulations are these figures from, the single-point case

or the 3D simulation? I would suggest more species are analyzed, covering a range of chemical lifetimes.

Response: The authors appreciate your comments. These figures are both from the single-point case, and we will conduct more experiments and analyze more species with diverse chemical lifetimes and properties.

Reference

Feng, F., Wang, Z., Li, J., and Carmichael, G. R.: A nonnegativity preserved efficient algorithm for atmospheric chemical kinetic equations, Applied Mathematics and Computation, 271, 519-531, https://doi.org/10.1016/j.amc.2015.09.033, 2015.

Feng, F., Chi, X., Wang, Z., Li, J., Jiang, J., and Yang, W.: A nonnegativity preserved efficient chemical solver applied to the air pollution forecast, Applied Mathematics & Computation, 314, 44-57, 2017.

Lawrence, B. N., Rezny, M., Budich, R., Bauer, P., Behrens, J., Carter, M., Deconinck, W., Ford, R., Maynard, C., Mullerworth, S., Osuna, C., Porter, A., Serradell, K., Valcke, S., Wedi, N., and Wilson, S.: Crossing the chasm: how to develop weather and climate models for next generation computers?, Geosci. Model Dev., 11, 1799-1821, 10.5194/gmd-11-1799-2018, 2018.

Xu, S., Huang, X., Oey, L. Y., Xu, F., Fu, H., Zhang, Y., and Yang, G.: POM.gpu-v1.0: a GPU-based Princeton Ocean Model, Geosci. Model Dev., 8, 2815-2827, 10.5194/gmd-8-2815-2015, 2015.

Please also note the supplement to this comment:
https://www.geosci-model-dev-discuss.net/gmd-2018-42/gmd-2018-42-AC2-supplement.pdf

---

## Author Response (AR1)

**Dear Editor,**

Thank you so much for extending our time for revision and we have already finished the revision of the manuscript. There are three important changes that we would like to inform you at first.

1. Following the comments by Referee #2, we coupled our MP CBM-Z to a regional chemistry transport model named the Nested Air Quality Prediction Model System (NAQPMS) model, and we presented the corresponding test results in section 3.3. We have added this part into the abstract and the conclusions.

2. We modified some re-write codes in previous version to eliminate or limit the error led by these re-write codes. To emphasize, we did not mean these modifications are wrong, but they would lead to the small difference of results of calculation. For example, the result of $x^2$ is not precisely equal to $x*x$ in Fortran since the low-level instructions are different for multiplication and 2-order exponentiation $x^2$. Therefore, we tried to keep these calculations were same for optimized and baseline codes. At the same time, some vectorization codes may lead to difference on results, so we stopped vectorization of some loops by adding directives. It will lead to some loss of performance but keep the consistence of results. Meanwhile, we also tested the codes under diverse compile flags to show the impacts of compile flags on performance and accuracy. The corresponding results were added to the revised manuscript.

3. We tested the codes on a new platform since the previous testing environment in Intel Cooperation was not available. Now we used the platform provided by Institute of Atmospheric Physics (IAP), Chinese Academic of Science (CAS). This computation cluster has nodes equipped with CPUs of Broadwell, Skylake architectures and KNL chips, which enables us to test and compare our optimization on diverse processors.

The final responses to the comments are presented as following.

**To Referee #1:**

**General comment**

In this manuscript, the authors design an architecture for CBM-Z chemical mechanism on next generation processors. This is interesting and quite useful for the routine numerical air quality forecast. I believe that this acceleration is helpful to policy managers. The acceleration of chemical solver is a difficult problem since 1980s. The computer technique has a rapid development. However, air quality models do not fully utilize this development. The contribution is generally well-written and complete. I suggested this manuscript to be published after considering the following comments.

Response: The authors thank for your time and your encouragement for this manuscript. As the reviewer mentioned, the hardware of computer has developed and improved, but the air quality model has not fully taken advantage of it to improve its computation efficiency, which may limit its practical value. This work is a part of the project about improving the computation performance of Global Nested Air Quality Prediction Model System (GNAQPMS) (Chen et al., 2015;Wang et al., 2017) and exploring the potential of the Intel Many Integrated Core (MIC) Architecture Chips on air quality model. We hope this work could be a good example to help other users to improve

the calculation efficiency of their models on the new generation chips. The detailed responses to the specific comments are given point to point as following.

**Specific Comments**

1. In figure 4, The authors plot the intercomparison of SO2. O3, H2O2, NO, H2SO4 between base and optimized simulations. I suggested some short-lived species like OH, HO2, RO2 should be compared, because these species is more sensitive to the mechinism, and very important to atmospheric oxdiation.

Response: The authors appreciate your constructive comments. We fully agree with your suggestion and has added the comparison of short-lived species like OH, HO2, RO2 in Figure 4 of the revised manuscript to further verify the model results. The Figure 4 is showed as following:

[Figure]

Figure 4. Comparison of the time-series concentrations of $O_3$, NO, $NO_2$, $H_2O_2$, $SO_2$, OH, $HO_2$, $RO_2$ and $H_2SO_4$ ((a)-(i)) from the baseline and optimized CBM-Z simulation with diverse -fp-model options.

2. The scenario in this manuscript is urban/polluted conditions. The authors present comparisons in other scenario like marine, biomass burning.

Response: The authors appreciate your important comments and accept your advice. In the revised paper, we used a three-dimension (3D) CTM coupled with MP CBM-Z to test the performance and validate the results. The test case for 3D CTM covers the majority of East Asia with 15km horizontal resolution, and it includes the scenario mentioned by the reviewer. We have present corresponding results in section 3.3. of the revised manuscript.

**To Referee #2:**

This paper describes improvements in vectorisation of the Carbon Bond Mechanism Z. While interesting, significant problems need to be addressed before I can recommend its publication in GMD.

Response: The authors thank you for encourage and sharing your time in this manuscript. We have merged your two files of comments and the detailed responses to the comments are given point to point.

**General Comments**

1. The manuscript contains numerous grammatical and spelling mistakes, and I will attempt to highlight these in detail in my specific comments below. I would recommend the authors check any resubmission carefully.

Response: The authors appreciate your comments. We have modified these grammatical and spelling mistakes of the current manuscript following your guides, and we will check the revised manuscript carefully before the resubmission.

2. Throughout the manuscript the authors make reference to "the chemistry transport model", "the CTM", and "the air quality model" without first describing which specific CTM or AQ model they are referring to, if any. I believe from the context that the model the authors use is in fact the GNAQPMS model, first mentioned on P2L13. If the authors mean a specific CTM it should be discussed in this context, although in some places (e.g. P3L15) the authors mean CTMs in general. If the authors mean any CTM then the phrase "Chemistry Transport Models" or "CTMs" would be appropriate.

Response: The authors agree with your comments. We would follow your advice and use the name of the GNAQPMS model when we referring to the specific model, meanwhile, we would use the phrase "Chemistry Transport Models" or "CTMs" when referring to CTMs in general.

3. The paper itself only covers improvements to the CBM-Z module, which I assume is included in some way into GNAQPMS, although this is not discussed by the authors. The CBM-Z code is provided by the authors on Zenodo, which is great to see, although I find it difficult to understand the improvements made in the code by examining Figures 2 and 3 in the manuscript. While graphical representations of these optimisations are useful, I would like to see how these were implemented in practice by the authors giving specific code examples within the manuscript.

Response: The authors appreciate your comments. Since we only focused on the CBM-Z module and isolate the impact of other modules in this manuscript, we didn't show the results of GNAQPMS model and only discussed CBM-Z module. However, the referee has mentioned that to show the improvement of MP CBM-Z in a CTM may has a more practical meaning. Therefore, we provided the tests results of Nested Air Quality Prediction Model System (NAQPMS) equipped with MP CBM-Z in revised paper. More details can be found in Section 3.3 in revised paper. In addition, we have followed referee's advice and added the sample codes to illustrate the implement of codes in the Figure 2 and Figure 3 of the revised paper. Figure 2 and Figure 3 in this response material is showed as following with sample codes:

[Figure]

Figure 2. A schematic diagram of the changes of the calling method of CBM-Z. The calling method of the CBM-Z module changes from calculating one model grid calculation at a time to multiple model grids at the same time. The VLEN represents the number of points operated simultaneously, which is determined by the length of the register in the Vector Processing Unit (VPU). The i and j loops, equaled latitude and longitude loops, were merged to construct one vector to reduce the number of unfilled vectors. (b) and (c) illustrate the sample codes before and after integrating grids.

[Figure]

Figure 3. The flowchart (a) shows the way to mask the heterogeneous girds to integrate grids to perform the vectorization operations according to the iregime values. (b) and (c) illustrate the sample codes before and after integrating grids. In figure (b), iregime leads different calling processes; in figure (c), the calling processes are integrated in one flow, and the function are called for all grids but only the values of valid grids would be returned.

4. The authors only give results from a CBM-Z standalone model, rather than having incorporated these improvements back into their CTM and presenting results from there. If available, I would certainly like to see what this does to model performance, as I feel it would strengthen this work greatly. As it is, they present only two case studies, where emissions are zero and meteorological conditions were constant. I would expect to see simulations of a number of different environments similar to those seen in simulations, i.e. free troposphere, boundary layer, urban, rural etc.

Response: The authors thank for your comments. We appreciate and accept your advices. To isolate contribution of CBM-Z to the performance of model, we didn't use the optimized GNAQPMS model, which has equipped with other optimizations. Instead, we used a pure regional CTM named NAQPMS to illustrate the performance improved of our optimizations on CBM-Z. In the CTM tests, we only replaced the original CBM-Z in NAQPMS with MP CBM-Z, and the description about NAQPMS in the revised paper is showed in Section 3.3.1 as following:

"NAQPMS is a regional CTM developed by IAP, CAS (Li et al., 2011;Li et al., 2013), and has been widely used in air quality research (Wang et al., 2018) and routine air quality forecasting (Wu et al., 2010;Chen et al., 2013). NAQPMS involves all essential processes including diffusion, advection, dry and wet deposition, multi-phase chemistry reactions. More details can be found in Li et al. (2013). Same as the box model, NAQPMS model with baseline and optimized CBM-Z were compiled with diverse compiling flags shown in Table 4."

The test case for CTM tests is a 72h simulation case covered the East Asia with 15-km horizontal resolution. The simulation started at 00:00 UTC, August 17, 2015 and ended in 00:00 UTC, August 20, 2015. We believe this test case could represent multiple scenarios because of its wide covering region. The description of test case in the revised paper is as:

"The test case is a 72-h simulation covered East Asia region. The horizontal resolution of grids is 15 km. The model adopted 20 vertical layers. The meteorological fields driven the NAQPMS model were provided by the Weather Research and Forecasting (WRF) model (Skamarock et al., 2008). The anthropogenic emission inventory was from the Hemispheric Transport of Air Pollution (HTAP) V2 and the organic emission inventory was provided by results from Sindelarova et al. (2014) using the Model of Emissions of Gases and Aerosols from Nature (MEGAN) (Guenther et al., 2006;Guenther et al., 2012). The simulation started at 00:00 UTC, August 17, 2015 and ended in 00:00 UTC, August 20, 2015."

5. CBM-Z output is plotted for 10 model hours, and within this time the relative error introduced by the optimisations is less than 0.05% (just), as seen in Figure 4 and mentioned throughout the manuscript. However, it is clear that this error is increasing for some species (e.g. H2O2, SO2, and H2SO4 as presented, and possibly others not shown). If these simulations were run for longer than 10 hours, would these errors still be below 0.05%? For confidence in the improvements described I would expect to see that the errors remain low for the length of a typical CTM simulation, which could be years depending on how the author's CTM is used. If the conditions used are more realistic (see point 4 above) do these errors increase?

Response: The authors appreciate your constructive comments. We agree with your comments. Indeed, the error of some species is increasing with time, and we also concern about this issue. Therefore, we have made some efforts to investigate the source of the error by conducting more tests and tried to find some solutions like using more reliable compiling flags to constrain the error. As showed in Figure 4 in the revised paper, adopting the -fp-model *precise* could decrease the error, however, it can't completely diminish the error because of the limitation of the single floating-precision calculation.

[Figure]

Figure 4. Comparison of the time-series concentrations of O3, NO, NO2, H2O2, SO2, OH, HO2, RO2 and H2SO4 ((a)-(i)) from the baseline and optimized CBM-Z simulation with diverse -fp-model options.

We followed the referee's comments and used a CTM to investigate the error of results. We did the comparison of model results with different version codes and different compile flags for accuracy controlling. Meanwhile, we chose some common gas-phase pollutants includes $NO_2$, $O_3$, $SO_2$ and CO and mainly compared the near surface results after a 72-h simulation or integration. Figure R3 and R4 shows the horizonal distribution of these pollutants and their absolute errors in near surface layer of model after 72-h simulation. They spatial distributions are generally same for all experiments and some points may exist the absolute errors. Figure R5 presents the distribution of absolute errors and relative errors among these grids.

[Figure]

Figure 5. NO2 and O3 concentrations outputted by baseline and optimized codes with different accuracy compile flags. (a) and (h) are from baselines codes compiled by precise option, which are treated as benchmark for comparison. (b) and (i) are from optimized codes compiled by precise option. (c) and (j) are from basline codse compiled by fast=1 flag. (d) and (k) are from optimized codes compiled by fast=1 flag. (e)-(g) and (l)-(m) are the output concentration differences of optimized codes (precise), baseline codes (fast=1) and optimized codes (fast=1) compared with baseline codes (precise).

[Figure]

Figure 6. SO2 and CO concentrations outputted by baseline and optimized codes with different accuracy compile flags. (a) and (h) are from baselines codes compiled by precise option, which are treated as benchmark for comparison. (b) and (i) are from optimized codes compiled by precise option. (c) and (j) are from basline codse compiled by fast=1 flag. (d) and (k) are from optimized codes compiled by fast=1 flag. (e)-(g) and (l)-(m) are the output concentration differences of optimized codes (precise), baseline codes (fast=1) and optimized codes (fast=1) compared with baseline codes (precise).

[Figure]

Figure 7. The distributions of Absolute Errors and Relative Errors for $O_3$, $NO_2$, $SO_2$ and CO in near surface model layer. The reference points are 1%, 25%, 50%, 75% and 99%.

The description of "Results Validation of CTM" can be found in Section 3.3.2 in the revised paper as:

"We chose four major gas pollutants includes NO2, O3, SO2 and CO after 72h integration or simulation at 00:00 UTC, August 20, 2015 to valid the optimized codes. The simulation results of old NAQPMS codes compiled by precise flag were as the benchmark results, and we mainly compared the simulation results of old NAQPMS codes with fast=1 flag and optimized NAQPMS with precise and fast=1, respectively.

Figure 5 and Figure 6 present the spatial distributions of NO2, O3, SO2 and CO as well as the absolute errors (AEs) of concentration of these species from other experiments. We can find that all model results show same reasonable spatial distribution of pollutants. In general, for NO2, O3 and SO2, the AEs of majority grids are in the range of ± 0.02 ppbv of other three experiments; for CO, the AEs of old and optimized NAQPMS with same fast=1 out that range shows the more obvious AEs.

The precise options enable the results of two version codes to be more consistent. Figure 7 shows the distribution of AEs and relative erros (REs) for four species in near surface model layer. For majority points, the AEs and REs are in a relatively small range. However, some points existed the exceptional obvious errors. The maximum AEs for NO2, O3, SO2 and CO are 0.166, 0.197, 0.001 and 0.03 ppb over the whole map after 72 integration. However, the fast=1 option shows more obvious error for both version codes. To old NAQPMS codes, using fast=1 leads to maximum 0.23, 4.5, 0.17 and 2.6 ppbv AEs for NO2, O3, SO2 and CO. To NAQPMS with MP CBM-Z, using the fast=1 option leads to maximum 0.13, 0.93, 0.76 and 0.64 ppbv AEs for NO2, O3, SO2 and CO over the whole map, which is relative better than the old NAQPMS.

Besides of considering the accuracy mentioned above, the determination of -fp-model option should also takes its impact on the performance in to account. In some pragmatic applications like routine air quality prediction, in an acceptable range, sacrificing the accuracy to gain is more reasonable. Conversely, the applications like long term

climate simulation, choosing more value-safe compile flags or adopting double-precision for calculation should be required to avoid accumulation of errors."

6. The authors describe running the CBM-Z model using a single point and over a number of grids for testing, which I believe to be a spatial grid from the context. In this case, is there any transport or mixing between grid-points, or any differences in meteorological variables? If not, how was this configured and set-up is each point solving the same conditions at all times? If so, this will not be representative of real-world usage where there will be a large amount of variation across the domain. For the single-point model, is it integrated in time? If not, how is it configured? I again question whether it would be better to perform these 3D simulations with a CTM instead of a stand-alone CBM-Z model, as it would then allow for a better understanding of how these improvements impact their model in practice. If this is not possible, the authors should give reasons as to why this is.

Response: The authors appreciate your comments. This case only indicates the idea performance of codes. The meteorological variables, e.g. temperature, are varied with grids in the multiple grid test. There is also no transport as well as diffusion but only gas-chemistry process in that test to isolate the impact of other processes.

To supplement the referee's concern, we presented the results of CTM tests in the revised paper. We used two version codes compiled with two compile flags for accuracy and tested on diverse platforms to see the improvement of codes as well as processors. The compile flags for vectorization and floating-point accuracy are shown in Table 3 in the revised paper.

**Table 3. Compile flags of the different versions of NAQPMS**

| Version of NAQPMS | Processor | Intel Compiler Flags | |
|---|---|---|---|
| | | Flags for Vectorization | Flags for Floating-point Accuracy |
| Baseline NAQPMS | **Xeon E5-2680 V4** | –xCORE-AVX2 | -fp-model precise |
| | | –xCORE-AVX2 | -fp-model fast=1 |
| | **Xeon Gold 6132** | –xCOMMON-AVX512 | -fp-model precise |
| | | –xCOMMON-AVX512 | -fp-model fast=1 |
| NAQPMS with MP CBM-Z | **Xeon E5-2680 V4** | –xCORE-AVX2 | -fp-model precise |
| | | –xCORE-AVX2 | –fp-model fast=1 |
| | **Xeon Gold 6132** | –xCOMMON-AVX512 | –fp-model precise |
| | | –xCOMMON-AVX512 | –fp-model fast=1 |
| | **Xeon Phi 7250** | –xMIC-AVX512 | –fp-model fast=1 |

The performance of old NAQPMS with precise was as the benchmark for comparison with other tests. All experiments were repeated 5 times to calculate the average performance. The performance results of different experiments are shown in Table 4.

Table 4. The performance tests of the baseline and optimized codes on the diverse platforms with different compile flags. The unit of the wall-times for the tests is second (s).

| | Processor | Vector Instruction | -fp-model | Wall-Time (CBMZ) | Wall-Time (Total) | Speedup (CBMZ) | Speedup (Total) |
|---|---|---|---|---|---|---|---|
| Baseline NAQPMS | Xeon E5-2680 V4 | AVX2 | precise | 17675.86 | 24460.54 | 1.00 | 1.00 |
| | | | fast=1 | 13201.56 | 19619.2 | 1.34 | 1.25 |
| | Xeon Gold 6132 | AVX512 | precise | 13817.24 | 18950.95 | 1.28 | 1.29 |
| | | | fast=1 | 10544.6 | 15502.39 | 1.68 | 1.58 |
| Optimized NAQPMS | Xeon E5-2680 V4 | AVX2 | precise | 11127.9 | 17454.95 | 1.59 | 1.40 |
| | | | fast=1 | 3971.48 | 10019.21 | 4.45 | 2.44 |
| | Xeon Gold 6132 | AVX512 | precise | 9584.59 | 14698.38 | 1.84 | 1.66 |
| | | | fast=1 | 2150.2 | 6994.43 | 8.22 | 3.50 |
| | Xeon Phi 7250 | AVX512 | fast=1 | 2997.96 | 19239.2 | 5.90 | 1.27 |

The description of "Performance tests of CTM" can be found in Section 3.3.3 in the revised paper as:

"The performance tests of old NAQPMS illustrate the improvement of performance led by the platform update and changing the compile flags. The performance of old NAQPMS with precise was as the benchmark for comparison with other tests. As showed in Table 6, in the original version NAQPMS, the CBM-Z module accounts for 72.26% wall-time of whole simulation. Changing the compile option of -fp-model to trade performance by sacrificing accuracy would lead to 1.34x and 1.25x speedups for the module CBM-Z and the whole model on Intel Xeon E5-2680 platform, respectively. By updating the CPU from Intel Xeon E5-2680 to Intel Xeon Gold 6132, the module CBM-Z and whole model gains 1.28x and 1.29x speedups, respectively. The speedups incline to 1.68x and 1.58x for CBM-Z and the whole model, respectively, by using fast=1 compile flag on Xeon Gold 6132 platform. The benefit from updating hardware is limited with the old codes, which implying the need of optimizing codes to adapt to the new hardware features.

The performances of gas-phase chemistry module and the NAQPMS model are largely improved after adopting MP CBM-Z described in this paper. As showed in Table 6, the CBM-Z model and the whole NAQPMS model gets 1.59x and 1.40x speedups on the old Xeon E5-2680 platform with same precise compile flag, and the speedups are improved to 4.45x and 2.44x by using the fast=1 compile flag. With same fast=1 flag, MP CBM-Z showed 3.32 and 1.96 times acceleration compared with old CBM-Z for gas-phase chemistry module and whole NAQPMS model. Such results illustrate that the optimization for vectorization releases the potential of existing hardware, the performance is highly improved even with the relative strict precise compile flag, which prevents most of vectorizations.

The new generation CPU further enforced the performance of MP CBM-Z. Using the platform with new generation processor Xeon Gold 6132, the speedups could reach 1.84x and 1.66x for the CBM-Z and the NAQPMS model with precise compile flag, respectively, and adopting fast=1 compile flag improves the speedups to 8.22x and 3.50x compared with benchmark performance. On the same Xeon Gold 6132 platform with fast=1 compile flag, MP CBM-Z gains 3.32 and 2.22 times acceleration compared with old CBM-Z for gas-phase chemistry module and whole NAQPMS model. Moreover, the time-consuming proportion of gas-phase chemistry declined to 30.74%, which is largely lower than that of 72.26% in baseline version model.

In addition, MP CBM-Z extended the benefit gained from advanced hardware. Using the same fast=1 compile option, the performance of old CBM-Z on AVX-512 platform is about 1.25 times of that on AVX-2 platform, and the performance of MP CBM-Z is about 1.84 times of that on AVX-2 platform. The efficiency of using new CPUs improved about 47% by adopting MP CBM-Z. Therefore, enhancing the vectorization ability of codes ensures that the applications, like CTM in this paper, could further utilize the improvement of processors on vectorization in the future.

KNL are more relied on SIMD to improve its performance according to the test results. The CBM-Z module is accelerated on KNL with a speedup of 5.9x, but the whole model only got a 1.27 times acceleration compared with benchmark performance. Comparing old CBM-Z on Intel Xeon Gold 6132 platform, MP CBM-Z gets a speedup of 3.52x for gas-phase chemistry on KNL, however, the performance of whole model declined 24%. Therefore, MP CBM-Z large improved the efficiency of CBM-Z on KNL by improving its vectorization ability, but further optimizations were required to let the whole CTM adapt to the architecture of KNL."

The conclusions about this part is added in the revised paper as:
"The tests using the single core showed that the vectorization optimization led to 5.16x and 8.97x speedups on Intel Xeon E5-2680 V4 and Intel Xeon Gold 6132 CPUs, respectively, and KNL gets a speedup of 3.69x comparing with performance of old CBM-Z on Intel Xeon E5-2680 V4 platform. It highlights the importance of vectorization to the KNL platform. Meanwhile, we also tested the MPI and OpenMP version CBM-Z. For the node, the speedup on the two generations CPUs can reach 104.63x and 198.50x using Message Passing Interface (MPI) and 101.02x and 194.60x using OpenMP, respectively, and the speedup on the KNL node can reach 194.60x using MPI and 167.45x using OpenMP. The speedup on the KNL node can reach 194.60x using MPI and 167.45x using OpenMP. The speedup of the optimized CBM-Z is approximately 40% higher on a 1-socket KNL platform than on a 2-socket Broadwell platform and about 13~16% lower than on a 2-socket Skylake platform.
The regional CTM NAQPMS were used to test the practical improvement of the MP CBM-Z in the real situation. The old and optimized codes of NAQPMS compiled with precise and fast=1 options, respectively, were tested the on diverse platforms. The outputs of models after 72-h simulation were used to validate and present the error by the codes as well as compile flags. The difference between old and optimized codes are generally in the range of ± 0.02 ppbv using precise option. The maximum over the whole map is about 0.166, 0.197, 0.001 and 0.03 ppbv for NO2, O3, SO2 and CO. The fast=1 option leads to more obvious error; however, performance could benefit a lot through adopting this option.
The choice of -fp-model compile flag decides the balance between the accuracy and performance. According to our test, after using the fast=1 option, the performance of codes would largely be improved but sacrificing some accuracy even using same codes. However, the loss of accuracy is relatively small, and in some practical applications that do not require high accuracy of floating-point calculation, it's acceptable to using fast=1 option to using the codes."

My major concerns with the manuscript as presented are:

A. Insufficient conditions used for testing

Response: We fully understand your concern and we has used the CTM with the optimized CBM-Z scheme to test the performance and validate the output of the model in a real scenario. As we mentioned above, corresponding descriptions and results have been added into the revised manuscript as Section 3.3.

B. Insufficient analysis of the errors introduced by the optimisations.

Response: In the current manuscript, we have stated that using -O0 compiling flag could diminish the difference of the results, which demonstrate there is no logical and artificial errors of the optimized codes. In the revised manuscript, we validated the results of models compiled by different compile flags for accuracy and performance control. It can't completely diminish the error because of the limitation of the single floating-precision calculation, therefore, we tried diverse combinations of codes and compile flags to discuss the balance between performance and accuracy. We suggest that we can use these optimizations to trade performance in some applications that didn't require the completely same results, for example, routinely air-quality prediction.

**Specific Comments and Technical Corrections**

P1L11 - computational

Response: We have accepted your advice and modified in the revised manuscript as following:

"Precise and rapid air quality simulation and forecasting are limited by the computational performance of the air quality model used, and the gas-phase chemistry module is the most time-consuming function in the air quality model."

P1L12 - model used.

Response: The authors appreciate your comments. We have accepted your advice and modified in the revised manuscript as following:

"Precise and rapid air quality simulation and forecasting are limited by the computational performance of the air quality model used, and the gas-phase chemistry module is the most time-consuming function in the air quality model."

P1L18 - Knights Landing

Response: We have accepted your advice and modified in the revised manuscript.

"The Intel Xeon E5-2680 V4 CPU and Intel Xeon Phi 7250 Knights Landing (KNL) are used as the benchmark processors."

P1L19 - I question whether the <0.05

Response: The authors appreciate your comments. The criteria of 0.05% comes from the results of single point test, and as mentioned above, we have further investigated the and discussed the source of errors by using CTM tests in the revised manuscript.

P1L29 - Chemistry Transport Models

Response: The authors appreciate your comments. We have accepted your advice and modified in the revised manuscript as following:

"As a useful tool for air quality problems, Chemistry Transport Models (CTMs) are widely used in studies of air quality"

P2L1 - a CTM

Response: The authors appreciate your comments. We have accepted your advice and modified in the revised manuscript as following:

"As the core of the AQF system, a CTM requires a large number of computational resources to simulate the complex chemical and physical processes."

P2L1 - computational

Response: The authors appreciate your comments. We have accepted your advice and modified in the revised manuscript.

"As the core of the AQF system, a CTM requires a large number of computational resources to simulate the complex chemical and physical processes."

P2L3 - relatively simple processes are adopted in CTMs to minimize

Response: The authors appreciate your comments. We have accepted your advice and modified in the revised manuscript as:

"To satisfy the demand of routine air quality forecasting in a timely manner, coarse spatial resolution and relatively simple processes are adopted in CTMs to minimize the use of computational resources."

P2L4 - computational

Response: The authors appreciate your precious comments. We have accepted your advice and modified in the revised manuscript as following

"To satisfy the demand of routine air quality forecasting in a timely manner, coarse spatial resolution and relatively simple processes are adopted in CTMs to minimize the use of computational resources."

P2L5-6 - I would suggest this: "Therefore, air quality simulation studies can benefit significantly by improving the performance of the CTM used."

Response: The authors appreciate your constructive suggestion. We have accepted your advice and modified in the revised manuscript as following:

"Therefore, air quality simulation studies can benefit significantly by improving the performance of the CTM used."

P2L16 - what is meant by "improving the frequency of air quality forecasting" in this context?

Response: The authors appreciate your comments. Yes, it means "improving the frequency of air quality forecasting" in the context and we have revised this sentence as following:

"The AQF system can also benefit from the performance improvement by adopting a higher model resolution and improving the frequency of air quality forecasting."

P2L19-21 - do you have any references for the trend in changes to computing architecture that can be quoted here?

Response: The authors appreciate your comments. Yes, Xu et al. (2015) and Lawrence et al. (2018) could be the appropriate reference here.

P3L4 - what is "the air quality model" in this context? Do you mean "in several air quality models"?

Response: The authors appreciate your comments. The "air quality model" here refers to the "CTMs". Following the second general comments of the reviewer, we would use the identical phase "CTMs" in the revised manuscript.

P3L9 - what do you mean by "architecture" in this context?

Response: The authors appreciate your comments. The phase "structure" may be more appropriate in this place, and it means we modified the codes structure of CBM-Z scheme, as we describe in the context, to improve the vectorization of codes.

P3L14 – tropospheric

Response: We have accepted your advice and modified in the revised manuscript as:

"CBM-Z is a lumped-structure photochemical mechanism that was developed to meet the needs of city-scale to global-scale tropospheric chemical simulation"

P3L15 - in CTMs

Response: We have accepted your advice and modified in the revised manuscript as following:

"The original scheme contains 67 species and 132 reactions. CBM-Z has been widely used in CTMs"

P4L1 - delete "The" and just start the sentence "CBM-Z also. . ."

Response: The authors appreciate your comments. We have accepted your advice and revised in the manuscript as:

"CBM-Z module still contains many scalar operations."

P4L6 - simulations

Response: The authors appreciate your comments. We have accepted your advice and modified in the revised manuscript as:

"Fortunately, contiguous model grids may have similar chemical processes in air quality simulations"

P4L14 - space after AVX-512

Response: The authors appreciate your comments. We have accepted your advice and modified in the revised manuscript.

P4L19 - do you mean "implement fine-grained parallelization"?

Response: The authors appreciate your comments. Yes, it is exactly what we mean and we have modified this part.

P4L21 - fine-grained

Response: The authors appreciate your comments, and we have revised in the manuscript as:

"Our goal is to implement fine-grained parallelization based on the SIMD and the grids that are distributed to a specific processor operate in parallel using the VPUs on each core."

P4L21 - delete the comma after SIMD

Response: The authors appreciate your comments. We have accepted your advice and modified in the revised manuscript.

P5L5 - does the solver use a fixed number of iterations, or does it integrate to convergence?

Response: The authors appreciate your comments. The solver uses the explicit algorithm and it didn't need iterations. The solver in this test is Modified-Backward-Euler (MBE) method. The detailed description of the solver could be found in Feng et al. (2015) and Feng et al. (2017).

P5L13-14 - does the fact that calculations are performed on all grids but not all grids are copied back introduce a possible inefficiency? Are these grids taking time to solve that could be better spent doing something else?

Response: The authors appreciate your comments. No, this process can keep the high computational efficiency of Vector Processing Union (VPU) considering that the chemistry processes among grids are different. It avoids the logical judgements to assort the grids so that corresponding computations can be finished on VPU simultaneously, which means that the computations of multiple grids could be finished at the same time without doing extra logical judgment processes.

P5L22-23 - as I have mentioned in the General Comments, how representative are these examples? How exactly were they configured?

Response: The authors appreciate your comments. As we mentioned in the manuscript, the goal of the single-point case is to validate the model's output for debugging logical or artificial errors, and the 3-dimension (3D) case is used for the performance testing. The 3D case is derived from the real CTM simulation, and the meteorological conditions like relative humility of this case is diverse among grids but constant during the simulation. The 3D case does not contain the emission, transport and diffusion process but only gas-chemistry process.

Following your advice, we would conduct some real-scenario simulation by using CTM. In addition, we will provide more clear descriptions of the test conditions as well as configuration in the revised manuscript.

P6L17 - as I have mentioned in the General Comments, how robust is the error of range <0.05

Response: The authors appreciate your comments. The criteria of 0.05% is based on our current test and we can't answer how robust of these criteria is without further test and investigation. We have further investigated the and discussed the source of errors by using CTM tests in the revised manuscript.

P7L4-9 - Please provide more details of these 3D simulations, as I am unclear exactly how they were set-up.

Response: The authors appreciate your comments. The 3-dimension case is derived from the real CTM simulation, and the meteorological conditions like relative humility of this case is diverse among grids but constant during the simulation. The 3D case does not contain the emission, transport and diffusion process but only gas-chemistry process.

P8L10 - It's great to see that the code is provided, but I found the structure provided confusing and the README provided lacking. However, this isn't part of this manuscript, but I would urge the authors to improve the documentation provided, include a directory listing with what the files do etc.

Response: The authors appreciate your comments. We would follow your advice to provide the codes with detailed document as well as directory structure for readers' or users' convenient.

P10 - Caption to Table 2: versions

Response: The authors appreciate your comments, and we have revised caption to Table 2 in the manuscript as:

"Compile flags of the different versions of CBM-Z."

P11 - Table 3: How many times were these tests run? Is it possible to provide an error estimate for these numbers?

Response: The authors appreciate your comments.

P12 - Caption to Figure 2: vectors

Response: The authors appreciate your comments, and we have accepted your advice in the revised manuscript as:

"The i and j loops, equaled latitude and longitude loops, were merged to construct one vector to reduce the number of unfilled vectors."

P12 - Figures 2: As mentioned in the General Comments, I believe that this manuscript would benefit from seeing how the code is altered with these optimizations.

Response: The authors appreciate your comments. We would provide the figures of sample codes to show the optimization processes.

P12 - Caption to Figure 3: grids

Response: The authors appreciate your comments and we have accepted your advice in the revised manuscript as:

"The flowchart shows the way to mask the heterogeneous girds to integrate grids to perform the vectorization operations according to the iregime values."

P13 - Figure 4: As mentioned in the General Comments, do these trends continue? How long until they become significant?

Response: The authors appreciate your comments. We are not sure about whether this error would continue. If these trends of error continue, it could become significant and effect the results. We also concern about this issue; therefore, we used a CTM to investigate the source of the error by conducting more tests and try to find some solutions. The results of corresponding tests and discussions can be found in the responses to general comment 4, 5 and 6.

P13/P14 - Figures 4 5: Which simulations are these figures from, the single-point case or the 3D simulation? I would suggest more species are analyzed, covering a range of chemical lifetimes.

Response: The authors appreciate your comments. These figures are both from the single-point case, and we will conduct more experiments and analyze more species with diverse chemical lifetimes and properties. We had used a CTM with MP CBM-Z to simulate a 72h case for comparing and analyzing the impact of optimization in the real scenario, and the results of corresponding tests and discussions can be found in the responses to general comment 4, 5 and 6.

**To Referee #3:**

**General comments**

This paper designed a new framework for an air quality model to adapt the new Intel architecture to improve the computing performance, which is very useful for HPC, model developer. I therefore fell that the manuscript is suitable for Geoscientific Model Development and Major Revision is needed. Some comments and suggestions follow below. I hope this paper can be more useful through further revisions.

Response: Thank you for your encourage and sharing the time in this manuscript. The detailed responses to the comments are given point to point as followed.

**Specific comments**

1. Writing

The writing is not good. I think the authors should look for a native language person to polish it. Some sentences are hard to understand, especially in 3.3 Performance Test and figure captions.

Response: The authors appreciate your precious comments. We have took some measures to improve the language of the transcript, especially the part in section 3.3 mentioned by the reviewer.

2. Experiments and analysis

  a. The authors did not give the information about the floating precision of the model. Double precision, or single precision? The performance should be difference, I think. If possible, the authors had better show the results of model with both single and double precision.

Response: Currently, the floating precision of the model is single-precision. We planned to do such tests, however, when we introduced the CTM to finish the tests, we found that testing double precision needed numerous adjustments of codes, including the re-statements of variables and adjustments of input datas. Therefore, we may consider doing the double precision in the future.

b. In this paper, the authors only give the results with one node. How about the model's captions? Is the performance (speedup) on KNL still better than CPU?

Response: The authors appreciate your constructive comments. As the reviewer mentioned, we only conducted the single node tests currently. This manuscript mainly focused on the vectorization of codes, which doesn't contribute to the factors which effects scalability of models. Meanwhile, this model is a box-model, and its scalability has no realistic value. Therefore, we did not do the tests to investigate the scalability in the revised paper. However, we provided the results of CTM tests in the revised paper, which may imply more practical meanings.

c. In performance test, authors only test the MPI and OpenMP separately. It's very interesting that the performances are almost same. Can authors explain that? I just wonder how about the computing performance by using MPI/OpenMP hybrid parallel method?

Response: The authors thank your for your precious comments. The MPI and OpenMP in our code were used for the same loop or parallel zone, therefore, the overhead of MPI was close to that of OpenMP in single node tests. Moreover, we only took the tests on single node, so, there is no overhead of MPI on message passing cross nodes through the network, which led to the similar performance of OpenMP and MPI. For the CBM-Z module, MPI and OpenMP are both used for the main loop currently, so it exists obstacle to realize the hybrid parallel method directly for this single loop, which requires more time and specific technological details. The hybrid parallel method could be a direction for our future work, and in this work, we would mainly focus on the exiting codes and results.

d. It takes 539.86s with one CPU core while 4481.10s with one KNL core. The authors blame the worse performance to lower frequency of KNL. It's difficult to understand because frequency difference between CPU and KNL is significantly and much less than the computing difference. Other factors, such as memory bandwidth, should also contribute the computing difference.

Response: The authors thank for your comments. As the reviewer mentioned, other factors like memory bandwidth

could also contribute to the gap of the performance. In our latest results, we noticed that vectorization has a obvious impact on code performance on KNL. If we used the compile flag to ensure accuracy and forbid the vectorization of many loops in optimized codes, the wall-time of KNL test inlined from 214.09 s to 3454.90 s. Under the architecture of KNL, vectorization plays a key role in performance.

**Reference**

Chen, H. S., Wang, Z. F., Li, J., Tang, X., Ge, B. Z., Wu, X. L., Wild, O., and Carmichael, G. R.: GNAQPMS-Hg v1.0, a global nested atmospheric mercury transport model: model description, evaluation and application to trans-boundary transport of Chinese anthropogenic emissions, Geosci. Model Dev., 8, 2857-2876, 10.5194/gmd-8-2857-2015, 2015.

Feng, F., Wang, Z., Li, J., and Carmichael, G. R.: A nonnegativity preserved efficient algorithm for atmospheric chemical kinetic equations, Applied Mathematics and Computation, 271, 519-531, https://doi.org/10.1016/j.amc.2015.09.033, 2015.

Feng, F., Chi, X., Wang, Z., Li, J., Jiang, J., and Yang, W.: A nonnegativity preserved efficient chemical solver applied to the air pollution forecast, Applied Mathematics & Computation, 314, 44-57, 2017.

Lawrence, B. N., Rezny, M., Budich, R., Bauer, P., Behrens, J., Carter, M., Deconinck, W., Ford, R., Maynard, C., Mullerworth, S., Osuna, C., Porter, A., Serradell, K., Valcke, S., Wedi, N., and Wilson, S.: Crossing the chasm: how to develop weather and climate models for next generation computers?, Geosci. Model Dev., 11, 1799-1821, 10.5194/gmd-11-1799-2018, 2018.

Wang, H., Chen, H., Wu, Q., Lin, J., Chen, X., Xie, X., Wang, R., Tang, X., and Wang, Z.: GNAQPMS v1.1: accelerating the Global Nested Air Quality Prediction Modeling System (GNAQPMS) on Intel Xeon Phi processors, Geosci. Model Dev., 10, 2891-2904, 10.5194/gmd-10-2891-2017, 2017.

Xu, S., Huang, X., Oey, L. Y., Xu, F., Fu, H., Zhang, Y., and Yang, G.: POM.gpu-v1.0: a GPU-based Princeton Ocean Model, Geosci. Model Dev., 8, 2815-2827, 10.5194/gmd-8-2815-2015, 2015.

---

## Author Response (AR2)

Dear Editor,

Thank you so much for your patience and efforts on our manuscript. We appreciated your meticulous guidance for improving the language of our paper. We have followed your advice and revised paper.

In addition, we also did three more revisions in the manuscript:

1. In page 9, line 11, we added the number of horizonal grids of test case in the description of test case as: "The horizontal resolution is 15 km with 339*432 grid boxes."

2. We added one more paragraph into the Conclusion Section about CTM test results as:

"The results of CTM test with the *fast=1* option show that the MP CBM-Z leads to a speedup of 3.32 and 1.96 for the gas-phase chemistry module and the CTM on Intel Xeon E5-2680 platform, respectively. Moreover, on the new Intel Xeon Gold 6132 platform, the MP CBM-Z gains 4.90x and 2.22x speedups for the gas-phase chemistry module and the whole CTM. For the KNL, the MP CBM-Z enables a 3.52x speedup for the gas-phase chemistry module, but the whole model lost 24.10% performance compared to the CPU platform due to the poor performance of other modules. Since this optimization seeks to improve the utilization of the VPU, the model is more suitable for the new generation processors adopting the more advanced SIMD technology. The results of our tests already show that the benefit of updating CPU improved by about 47% by using the MP CBM-Z since the optimized code has better adaptability for the new hardware. "

3. In the part (c) of figure 3, the original figure misses two lines codes. So we had revised this part by adding them back.

Some detailed changes are presented as follow.

Thank you for submitting a revised manuscript and for largely addressing the reviewers' comments. As one of the reviewers said, the quality of the language could be better and I include below a detailed list of corrections to address this. In future, I would strongly suggest that you seek advice on the language when drafting a paper; it will improve the readability of your papers and help the review process.

**Response**: Thank you so much for your advice. We really appreciated the reviewers' constructive comments and spent about four months addressing the reviewers' concerns. We would like to thank two reviewers and the editor for your patience during this long process.

We also sincerely express our gratitude to the editor for your guidance on improving our poor language of the manuscript. We have addressed all language issues you mentioned in the manuscript, and we would pay more attention to this aspect in the future. To save the space, we response all language issue comments here at one time, but for some specific comments, we would response them point to point as follow.

I ask that you implement these corrections. Also, it wasn't clear to me that you'd altered the Figure 4 caption in response to one of the reviewers. Can you please do so? Once you've done these changes, I'll be happy to recommend that your paper be accepted and published in GMD.

**Response**: Thank you so much for your comments. We appreciate your hard works on our manuscript. We have added more description in the caption of Figure 4 as you asked:

"Figure 1. Comparison of the time-series concentrations of $O_3$, NO, $NO_2$, $H_2O_2$, $SO_2$, OH, $HO_2$, $RO_2$ and $H_2SO_4$ ((a)-(i)) from the baseline and optimized CBM-Z simulation with diverse -fp-model options. The simulation results by the baseline code with the ***-fp-model precise*** compile flag was as the benchmark. The solid lines show the time-series concentrations of the species from different experiments and the dashed lines showed the Relative Errors (RE) of simulated concentrations between the benchmark and the results by other combinations of the code and -fp-model options."

30. Method Description, page 3, line 15: I suggest that you change "too many branches and unbalanced calculations" to "too many options and poor load balancing" as readers unfamiliar with the structure of CBM-Z will not understand what it meant by "branches". It's also a word commonly used to add new code to a model that isn't included in the release and so, probably isn't appropriate until you've introduce CBM-Z in Section 2.1.

**Response**: We appreciate your precious comments. We have followed your advice and changed the corresponding line in the manuscript as:

"In this module, too many options and poor load balancing within the model grid boxes make it a challenge to improve its performance on a vectorization level."

42. Section 2.2, page 4, line 29: Change "the model grids" to "the model grid boxes". This requested change occurs on this line but elsewhere throughout section 2.2

**Response**: Thank you so much for your comments. We have changed "the model grids" to "the model grid boxes" in the line you mentioned and the rest part of manuscript.

47. Section 2.2, page 5, line 15: the line "The VLEN number grids contain …" is very unclear and I would ask that you re-write this with more clarity

**Response**: We appreciate your constructive comments. We have re-written this sentence as:

"A set of grid boxes with the number of VLEN (16 in this study) would perform the operation simultaneously, and the variable ***pmask*** signed the valid grid boxes."

91. Section 3.2, page 7, line 6: I felt that this line is repetitive and could be removed. However, the point about the meteorology seems to contradict what was stated in 3.2.1. Was the meteorology constant or updated?

**Response**: Thank you for your precious comments. We apologize for the mistake here since the meteorological conditions is constant for the single grid box simulation. We have followed your suggestion and removed this line from the manuscript.

173. Section 3.3, page 9, line 17: There are many places where you refer to the "old" CBM-Z. I suggest that you remove all references to "old" and replace with either "unoptimized" or "benchmark" or "baseline". Whichever you choose, use consistently throughout the manuscript

**Response**: Thank you for your constructive comments. We have replaced the word "old" with "baseline" in the manuscript. We also followed your suggestion and used "baseline" consistently throughout the whole manuscript.